# B-cos LM: Efficiently Transforming Pre-trained Language Models for Improved Explainability

**Yifan Wang**                                                              *yifwang@lst.uni-saarland.de*
*Saarland University, Saarbrücken, Germany*

**Sukrut Rao**                                                             *sukrut.rao@mpi-inf.mpg.de*
*Max Planck Institute for Informatics, Saarland Informatics Campus, Saarbrücken, Germany*

**Ji-Ung Lee**                                                              *ji-ung.lee@uni-saarland.de*
*Saarland University, Saarbrücken, Germany*

**Mayank Jobanputra**                                                         *mayank@lst.uni-saarland.de*
*Saarland University, Saarbrücken, Germany*

**Vera Demberg**                                                              *vera@lst.uni-saarland.de*
*Saarland University, Saarbrücken, Germany*
*Max Planck Institute for Informatics, Saarland Informatics Campus, Saarbrücken, Germany*

**Reviewed on OpenReview:** *https://openreview.net/forum?id=c18OUH8Dg8*

## Abstract

Post-hoc explanation methods for black-box models often struggle with faithfulness and human interpretability due to the lack of explainability in current neural architectures. Meanwhile, B-cos networks have been introduced to improve model explainability by proposing an architecture that removes bias terms and promotes input-weight alignment. Although B-cos networks have shown success in building explainable systems, their application has so far been limited to computer vision models and their associated training pipelines. In this work, we introduce B-cos LMs, i.e., B-cos Language Models (LMs) empowered for natural language processing (NLP) tasks. Our approach directly transforms pre-trained language models into B-cos LMs by combining B-cos conversion and task fine-tuning, improving efficiency compared to previous methods. Automatic and human evaluation results demonstrate that B-cos LMs produce more faithful and human interpretable explanations than post-hoc methods, while maintaining task performance comparable to conventional fine-tuning. Our in-depth analysis explores how B-cos LMs differ from conventionally fine-tuned models in their learning processes and explanation patterns. Finally, we present a first exploration of transforming decoder-only models to B-cos LMs for generation tasks. Our code is available at https://github.com/Ewanwong/bcos_lm.

## 1 Introduction

Pre-trained language models (PLMs) such as BERT (Devlin et al., 2019) and GPT (Radford et al., 2019; Brown et al., 2020; OpenAI, 2023) have significantly advanced performance across a plethora of NLP tasks (Wang et al., 2018; Gao et al., 2023). However, their complex architectures and black-box nature make understanding their behavior a persistent challenge (Bommasani et al., 2021). To address this, research has increasingly focused on understanding model predictions in various natural language understanding and generation tasks using different forms of explanations, such as input-based explanations (Feng et al., 2024; Wei Jie et al., 2024; Jiang et al., 2024; Madsen et al., 2024; Yin & Neubig, 2022; Deiseroth et al., 2023; Abbasi et al., 2025), natural language explanations (Ramnath et al., 2024; Wang et al., 2025a), and concept-based explanations (Yu et al.,

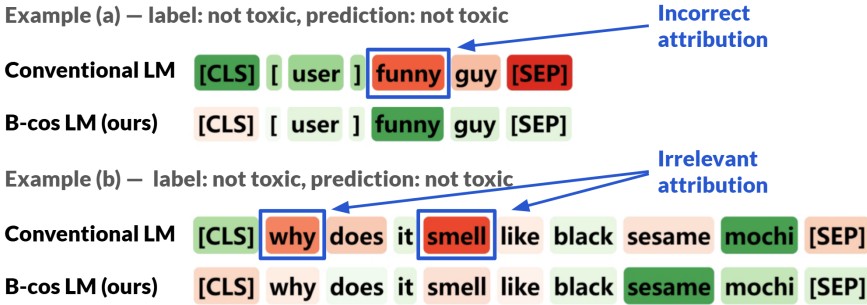

Figure 1: **Visualization of $\mathbf{W(x)x}$ in a conventionally fine-tuned model (Conventional LM) and a B-cos LM.** Green (red) indicates the positive (negative) impact of tokens on the prediction. In both examples, both models correctly predict *not toxic*. In the Conventional LM, "funny" is incorrectly assigned a negative attribution in example (a), and in example (b), irrelevant words like "why" and "smell" are highlighted, making the explanations unfaithful and less interpretable. Examples and explanations are drawn from HateXplain. See §3 for details on how $\mathbf{W(x)x}$ is computed.

2024; Raman et al., 2024). Among others, input-based explanations, often referred to as rationales, aim to reveal how specific inputs influence a model's prediction (Arras et al., 2019; Atanasova et al., 2020; Lyu et al., 2024). In this work, we focus on input-based explanations, as they offer the most direct insight into model behavior and are often mandated by laws, such as the EU Artificial Intelligence Act.

Most input-based explanation methods for neural models are post-hoc, meaning that they attempt to explain a model's behavior only after it has been trained and deployed (Lyu et al., 2024). While these methods are widely used and easy to apply, they have been shown to produce unfaithful explanations that do not accurately reflect the model's actual reasoning process (Kindermans et al., 2019; Slack et al., 2020; Pruthi et al., 2020; Ye et al., 2025). They also struggle with human interpretability, making it difficult for users to understand the model's reasoning (Smilkov et al., 2017; Ismail et al., 2021). Prior research suggests that these limitations stem from a lack of inherent explainability in current models, that is, the model's ability to generate faithful and interpretable explanations by design (Kindermans et al., 2018; Alvarez Melis & Jaakkola, 2018; Rudin, 2019). As a result, improving model explainability is crucial for producing explanations that are both reliable and useful to users.[1] Figure 1 provides examples illustrating this issue.

To overcome these limitations, we introduce **B-cos LM**, a dynamic linear model that learns the most task-relevant patterns through increased input-weight alignment pressure. Building upon B-cos networks that were first introduced by Böhle et al. (2022) for computer vision, we ensure the explainability of B-cos LMs through mathematically grounded architectural and computational adaptations, with specialized architectural modifications and training pipelines tailored for NLP tasks.

We conduct comprehensive empirical experiments using encoder-only models on classification tasks. Our focus on classification is motivated by its prevalence in high-stakes applications, such as loan approvals, hiring decisions, and hate speech detection, where explainability is crucial and often legally mandated. Encoder-only models have also seen renewed interest in the research community (Warner et al., 2024; Breton et al., 2025; Chaffin, 2025), and they remain the standard architecture for text classification tasks and continue to perform competitively compared to large language models (LLMs) (Zhao et al., 2024). Beyond that, we also explore applying B-cos LMs to decoder-only models for generation tasks and show that B-cos LMs can be extended to a variety of tasks and the latest model architectures. Our contributions are as follows:

1. We propose B-cos LM, a novel model with enhanced explainability. Automatic and human evaluations demonstrate that B-cos LMs generate more faithful and human interpretable explanations than post-hoc explanations while maintaining a strong task performance.
2. We investigate different strategies for transforming PLMs into task-specific B-cos LMs. Our findings show that combining task fine-tuning and B-cos conversion is the most efficient approach, leading to faster convergence than previous B-cos methods and conventional fine-tuning.

---

[1]Considering the evolving definition of these terms in past literature, we provide a detailed definition in Appendix A.

3. We thoroughly investigate how B-cos LMs differ from conventionally fine-tuned models and examine how alignment pressure influences their behavior.
4. We are also the first to explore the transformation of decoder-only models to B-cos LMs for generation tasks, providing a step towards a broader application of B-cos LMs in the era of LLMs.

## 2 Related Work

**Post-hoc Explanation Methods** Various methods have been proposed to provide post-hoc explanations for neural model predictions (Atanasova et al., 2020). These methods can be broadly categorized based on how they generate explanations: gradient-based (Simonyan et al., 2014; Kindermans et al., 2016; Sundararajan et al., 2017; Enguehard, 2023), propagation-based (Bach et al., 2015; Shrikumar et al., 2017; Springenberg et al., 2015; Ferrando et al., 2023; Modarressi et al., 2022; 2023), and perturbation-based methods (Li et al., 2016; Ribeiro et al., 2016; Lundberg & Lee, 2017; Deiseroth et al., 2023). Besides, the attention mechanism (Bahdanau et al., 2015) is often viewed as an explanation, particularly in transformer-based models (Vaswani et al., 2017). While most existing work focuses on understanding model predictions in classification settings, recent efforts have also aimed to explain model behavior in generation tasks, including sentence completion (Yin & Neubig, 2022; Ferrando et al., 2023), question answering (Enouen et al., 2024), and summarization (Cohen-Wang et al., 2024).

Although post-hoc methods have been widely used, numerous studies have shown that they lack faithfulness, often failing to capture the true decision-making process of the model (Kindermans et al., 2019; Jain & Wallace, 2019; Slack et al., 2020; Pruthi et al., 2020; Ye et al., 2025). Furthermore, they are noisy and may select irrelevant information, leading to explanations that cannot be interpreted by humans (Smilkov et al., 2017; Ismail et al., 2021).

**From Post-hoc Explanations to Explainable Models** Prior research suggests that the lack of faithfulness and human interpretability in post-hoc explanations arises from the fundamental lack of explainability in modern neural models, which are typically optimized solely for task performance (Kindermans et al., 2018; Rudin, 2019; Atanasova et al., 2022). In response, various efforts have been made to enhance model explainability. Some works have introduced constraints that improve specific explanation properties, such as faithfulness (Tutek & Šnajder, 2022; Moradi et al., 2020; 2021; Barkan et al., 2024), consistency (Atanasova et al., 2022), locality (Alvarez Melis & Jaakkola, 2018), and plausibility (Ismail et al., 2021). However, as these constraints are typically imposed as regularizers, their effectiveness in improving explanation quality is not guaranteed (Pruthi et al., 2020). Others have proposed self-explanatory model architectures such as rationale-based models that utilize an "explain-then-predict" pipeline, where one module selects rationales for another to make predictions based on them (Lei et al., 2016). Although seemingly transparent, both components rely on neural networks, making the rationale extraction and utilization processes opaque (Zheng et al., 2022; Jacovi & Goldberg, 2021). Besides, such models may face optimization challenges that limit their practicality in real-world tasks (Lyu et al., 2024).

To tackle these shortcomings, Böhle et al. (2022) proposed B-cos networks. Unlike methods that impose external constraints, B-cos networks improve explainability through mathematically grounded architectural and computational adaptations. Moreover, these adaptations are designed as drop-in replacements for conventional model components, making B-cos networks easy to train with minimal performance loss. Most recently, Arya et al. (2024) explored B-cosification techniques to convert existing models into B-cos models, which reduces the training costs of adopting B-cos architectures.

Despite their successful application in vision tasks, B-cos networks have yet to be explored in NLP, where input modalities and training paradigms differ significantly. In this work, we adapt B-cos models for the language domain, integrating them efficiently into NLP pipelines.

## 3 Methodology

In this section, we outline the architecture and training process of B-cos LMs and how their design ensures faithful and human interpretable explanations. We first introduce B-cos networks (§ 3.1) and then describe

how we transform PLMs to task-specific B-cos LMs (§ 3.2). Finally, we demonstrate how to generate explanations from B-cos LMs (§ 3.3). Notations used in the work are detailed in Appendix B.

## 3.1 B-cos Networks

Complex neural networks can be interpreted as generalized linear models (Nair & Hinton, 2010; Alvarez Melis & Jaakkola, 2018; Srinivas & Fleuret, 2019). For each input $\mathbf{x}$, the network effectively applies a linear transformation: $\mathbf{f}(\mathbf{x}) = \mathbf{W}(\mathbf{x})\mathbf{x} + \mathbf{b}(\mathbf{x})$, where both the weight $\mathbf{W}(\mathbf{x})$ and bias $\mathbf{b}(\mathbf{x})$ depend on $\mathbf{x}$. Given that many activation functions are (approximately) piecewise linear, the overall network can be viewed as (approximately) piecewise affine (Alvarez Melis & Jaakkola, 2018). Earlier work refers to such models as dynamic linear models (Böhle et al., 2021; 2022), highlighting the fact that the weight and bias terms dynamically change according to $\mathbf{x}$.

Under this dynamic linear perspective, the linear mapping $\mathbf{W}(\mathbf{x})$ can be seen as attributing model predictions to individual input features, and $\mathbf{W}(\mathbf{x_i})\mathbf{x_i}$ can be seen as the contribution of feature $\mathbf{x_i}$ to the model prediction. However, two challenges hinder the direct use of this interpretation. First, $\mathbf{W}(\mathbf{x})$ alone provides an incomplete and unfaithful model summary, since $\mathbf{f}(\mathbf{x}) \neq \mathbf{W}(\mathbf{x})\mathbf{x}$ due to the presence of the bias term $\mathbf{b}(\mathbf{x})$, and incorporating $\mathbf{b}(\mathbf{x})$ into explanations is highly non-trivial (Wang et al., 2019). Second, $\mathbf{W}(\mathbf{x_i})\mathbf{x_i}$ is often difficult for humans to interpret, as $\mathbf{W}(\mathbf{x})$ does not necessarily align only with task-relevant input patterns (Smilkov et al., 2017) and therefore yields noisy and irrelevant explanations. Figure 1 illustrates these challenges. To address these issues, Böhle et al. (2022) introduced B-cos networks by replacing the conventional linear transformation:

$$\mathbf{f}(\mathbf{x}; \mathbf{w}, \mathrm{b}) = \mathbf{w^T}\mathbf{x} + \mathrm{b} = \|\mathbf{w}\|\|\mathbf{x}\|\cos(\mathbf{x}, \mathbf{w}) + \mathrm{b} \tag{1}$$

with a B-cos transformation:

$$\text{B-cos}(\mathbf{x}; \mathbf{w}) = \hat{\mathbf{w}}^\mathbf{T}\mathbf{x} \times |\cos(\mathbf{x}, \hat{\mathbf{w}})|^{\text{B-1}} \tag{2}$$
$$= \|\hat{\mathbf{w}}\|\|\mathbf{x}\||\cos(\mathbf{x}, \hat{\mathbf{w}})|^{\text{B}} \times \text{sgn}(\cos(\mathbf{x}, \hat{\mathbf{w}}))$$

where $\hat{\mathbf{w}}$ is a scaled version of $\mathbf{w}$ with unit norm and sgn denotes the sign function.

$\text{B-cos}(\mathbf{x}; \mathbf{w})$ can be seen as a linear transformation of $\mathbf{x}$ with the dynamic linear weight $\mathbf{w}(\mathbf{x}) = |\cos(\mathbf{x}, \hat{\mathbf{w}})|^{\text{B-1}} \times \hat{\mathbf{w}}$. The absence of $\mathbf{b}(\mathbf{x})$ ensures the completeness of summary $\mathbf{w}(\mathbf{x})$. We show that all components of transformer models can be viewed as or easily converted to bias-free, dynamic linear modules in Appendix C, and demonstrate that this completeness extends to an entire network composed of such modules in § 3.3. Moreover, since the B-cos module output is bounded by $\|\mathbf{x}\|$, the weight $\mathbf{w}$ must align closely with task-relevant patterns to achieve a high cosine similarity and strong activation, especially under additional alignment pressure (B>1). This drives the model to assign greater weight to the most relevant features when optimizing target output probabilities, promoting the learning of representative patterns during training. Consequently, during explanation generation, task-relevant features $\mathbf{x_i}$ receive higher attribution $\mathbf{W}(\mathbf{x_i})\mathbf{x_i}$ due to stronger alignment, while irrelevant features receive lower attribution, suppressed by weaker alignment and the exponential scaling. For a more detailed discussion of how the B-cos transformation enhances faithfulness and human interpretability, please see Böhle et al. (2022; 2024).

While early B-cos models were trained from scratch, Arya et al. (2024) recently introduced B-cosification, an efficient method to obtain B-cos models. This approach first modifies conventional models with task capacities to adopt the B-cos architecture, followed by fine-tuning on downstream datasets for B-cos conversion. B-cosified models generate explanations as faithful and interpretable as B-cos models trained from scratch but at a much lower training cost. However, directly applying B-cosification to LMs is non-trivial and inefficient due to the significant differences in model architectures and training pipelines.

## 3.2 B-cosification for LMs

In this section, we present our B-cosification approach for LMs. We summarize the differences between B-cosification for LMs, its counterpart for vision models, and conventional fine-tuning in Table 1.

| Property | Conventional Fine-tuning | B-cosification for vision (Arya et al., 2024) | B-cos LM (ours) |
|---|---|---|---|
| **Bias terms** | yes | no | no |
| **B (alignment pressure)** | 1 | 2 | 1.25 / 1.5 |
| **Pred. Head Activations** | tanh | n/a[2] | identity |
| **Prior task abilities** | no | yes | no |
| **Training objectives** | Task fine-tuning | B-cos conversion | Task fine-tuning & B-cos conversion |

Table 1: Comparison between conventional fine-tuning, B-cosification for vision models and B-cosification for language models (B-cos LM). Conventional fine-tuning and B-cosification for vision follow the configuration of BERT for sequence classification and CLIP (Radford et al., 2021), respectively (cf. § 3 for details).

### 3.2.1 B-cos Adaptations

Given a conventional model, we first modify its architecture and computation to integrate the B-cos framework.

**Architectural Adaptations** For completeness and faithfulness of explanations, we follow Arya et al. (2024) and remove all bias terms in models, including those in the affine transformations of layer normalization and attention blocks. Additionally, a prediction head is typically added on top of transformers before fine-tuning for downstream tasks in the NLP pipeline. This head often includes activation functions that are not (approximately) piecewise linear, such as sigmoid and tanh. To accommodate the unique architecture of LMs, we remove all activation functions in the prediction heads, as they generate explanations that are not locally difference-bounded (Alvarez Melis & Jaakkola, 2018) and introduce numerical instability during explanation generation. Our experiments show that the added non-linearity from B>1 could compensate for this removal.

**Introducing B-cos Computation** To promote input-weight alignment and improve human interpretability of explanations, we replace all linear transformations with B-cos transformations in § 3.1. For a more efficient B-cosification, B-cos layers are initialized with the corresponding weights of the original model.

### 3.2.2 Fine-tuning

The B-cos adaptations above modify the architecture and computation of models, requiring fine-tuning to restore their capabilities and adapt to alignment pressure. Following the "pre-train then fine-tune" paradigm, which is frequently utilized in NLP tasks, we directly transform PLMs to B-cos LMs, rather than adapting task-specific models as done in previous work (Arya et al., 2024). This fundamental difference in the training pipeline adds complexity to B-cosification for LMs, as the objective involves both B-cos conversion and task fine-tuning. While there are multiple ways to conjoin these two steps (cf. § 5), we find that the most efficient way is to combine them by first applying B-cos adaptations to a PLM and then fine-tuning it on a downstream task. Following Böhle et al. (2022), we use the binary cross-entropy (BCE) loss instead of the conventional cross-entropy loss, as it explicitly maximizes the absolute target logits and strengthens the alignment pressure. We provide an extensive comparison of different B-cosification setups in § 5.

### 3.3 Computing B-cos Explanations

Once trained, the B-cos LM can generate explanations that faithfully summarize its decision-making process during inference. As all components are dynamic linear with no bias terms (cf. Appendix C), the entire model computation can be expressed as a sequence of matrix multiplications, which can be completely summarized as a single dynamic linear function:

$$\hat{\mathbf{W}}_L(\mathbf{A}_L)\hat{\mathbf{W}}_{L-1}(\mathbf{A}_{L-1})...\hat{\mathbf{W}}_1(\mathbf{A}_1 = \mathbf{X})\mathbf{X} = \Pi_{j=1}^{L}\hat{\mathbf{W}}_j(\mathbf{A}_j) \tag{3}$$

Note that a residual connection of $\mathbf{W}(\mathbf{x})\mathbf{x} + \mathbf{x}$ with $\mathbf{x} \in \mathbb{R}^n$ and $\mathbf{W}(\mathbf{x}) \in \mathbb{R}^{n \times n}$ is mathematically equivalent to a single dynamic linear transformation of $(\mathbf{W}(\mathbf{x}) + \mathbf{I}_n)\mathbf{x}$. Considering the textual inputs specific to LMs, we attribute the model's predictions to the embedding representations. Specifically, to quantify the contribution of a token $i$ to a model prediction, we compute the dot product $\mathbf{W}(\mathbf{x}_i)\mathbf{x}_i$ between its embedding $\mathbf{x}_i$ and the

---

[2]Arya et al. (2024) used a single linear layer on top of CLIP so the prediction head activation is not applicable in their setup.

corresponding dynamic linear weight $\mathbf{W}(\mathbf{x}_i)$ for the target class logit. For the remainder of the paper, we will refer to such explanations as *B-cos explanations.*

## 4    Experiments

We evaluate the task performance of B-cos LMs and faithfulness of B-cos explanations with automatic evaluation across various tasks and PLMs. In addition, we conduct a human evaluation study to compare the human interpretability of B-cos explanations. §4.1–4.3 describe our automatic evaluation setup, results, as well as human evaluation study, respectively. §4.4 provides a qualitative analysis. Finally, we conduct an ablation study in §4.5. More details on the experimental setup and baseline methods are provided in Appendix D and a comparison of computational efficiency is provided in Appendix E.

### 4.1    Experimental Setup

**Datasets and Models**    Our experiments use three datasets: AG News (topic classification, Zhang et al., 2015), IMDB (sentiment analysis, Maas et al., 2011), and HateXplain (hate speech detection, Mathew et al., 2021). BERT (Devlin et al., 2019), DistilBERT (Sanh et al., 2019), and RoBERTa (Liu et al., 2019) are used as the basis for conventional fine-tuning and for obtaining B-cos LMs. We set B=1.25 for IMDB and B=1.5 for AG News and HateXplain datasets.

**Baselines**    We compare B-cos explanations against a diverse set of post-hoc explanation methods: Attention (Bahdanau et al., 2015), InputXGradient (IxG, Kindermans et al., 2016), Sequential Integrated Gradients (SIG, Enguehard, 2023), DecompX (Modarressi et al., 2023), Shapley Value Sampling (ShapSampl, Strumbelj & Kononenko, 2010), and LIME (Ribeiro et al., 2016). We also apply these methods to a model trained with Saloss (Chrysostomou & Aletras, 2021), which introduces faithfulness regularization. This setup enables a direct comparison between B-cos LMs and models specifically optimized for explainability. For embedding-level explanation methods, we aggregate attributions by summing across all embedding dimensions.

**Faithfulness Metrics**    For a more comprehensive evaluation, we employ two different methods to assess faithfulness. First, we report two perturbation-based metrics (DeYoung et al., 2020):

- **Comprehensiveness** (Comp) measures the average drop in predicted class probability after masking out the top $k\%$ most important tokens in the explanation. A higher score indicates better faithfulness.
- **Sufficiency** (Suff) measures the average drop in predicted class probability after keeping only the top $k\%$ tokens. A lower score indicates better faithfulness.

To avoid arbitrary choices of $k$, we compute Comp and Suff for multiple values ($k = 10, 20, ..., 90$) and summarize them using the Area Over the Perturbation Curve (AOPC, DeYoung et al., 2020).

In addition, we introduce a new faithfulness metric called Sequence Pointing Game (SeqPG), inspired by the grid pointing game in vision tasks (Böhle et al., 2021):

- **Sequence Pointing Game** (SeqPG). We evaluate models on synthetic sequences composed of segments associated with different classes. To assess faithfulness, we measure the proportion of positive attribution assigned to the corresponding segment of each class and compute their average. A higher score indicates better faithfulness.

Compared to perturbation-based metrics, SeqPG does not rely on perturbations and thus avoids the potential distortions introduced by token masking. When constructing SeqPG examples, we truncate each segment to a fixed length and randomize segment order to control for length and position effects. We generate synthetic examples using correctly and most confidently classified test instances. SeqPG can be seen as a standardized version of hybrid document evaluation (Poerner et al., 2018). We provide an example of SeqPG in Figure 7 and more details in Appendix F.

## 4.2 Automatic Evaluation Results

**Task Performance** Figure 2 shows the accuracy of conventionally fine-tuned, Saloss and B-cos BERT across three datasets (we provide results for DistilBERT and RoBERTa in Appendix G). On AG News and HateXplain, B-cos LMs performs on par with conventional models, with only a minor drop (∼1%) in accuracy. They also outperform Saloss models on these datasets. Only for IMDB, we find a slightly larger drop of 3.06% compared to conventional BERT, though the performance remains strong overall.

**Faithfulness Results** Table 2 shows the faithfulness scores for post-hoc explanation methods on conventionally fine-tuned and Saloss BERT models, as well as B-cos explanations from B-cos BERT. The results show that B-cos explanations are consistently and substantially more faithful than post-hoc methods across all models and datasets. On average, B-cos explanations outperform the strongest post-hoc methods on conventional models by 14.63 points in Comp and achieve negative Suff scores, indicating that the identified important tokens alone enable even more confident predictions. B-cos also shows significant gains in SeqPG. While Saloss improves faithfulness for

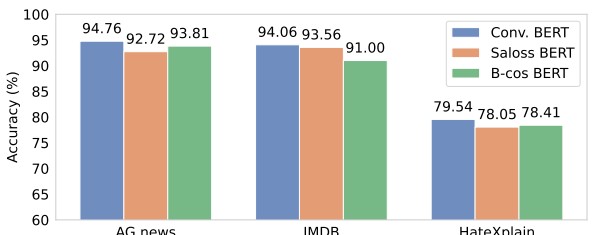

Figure 2: Mean accuracy of conventionally fine-tuned, Saloss and B-cos BERT averaged over three runs. B-cos models perform comparably to conventional models on most tasks.

some post-hoc methods over conventional models, it still lags markedly behind B-cos LMs. Similar trends are observed for other PLMs (Appendix H) as well. Although we do not include rationale-based models in the main experiments because they typically require additional supervision, a supplementary comparison in Appendix I shows that B-cos BERT still outperforms a rationale-based model on HateXplain.

| Model | Method | AG News | | | IMDB | | | HateXplain | | |
|---|---|---|---|---|---|---|---|---|---|---|
| | | Comp (↑) | Suff (↓) | SeqPG (↑) | Comp (↑) | Suff (↓) | SeqPG (↑) | Comp (↑) | Suff (↓) | SeqPG (↑) |
| Conv. BERT | Attention | 24.40 | 8.09 | 50 | 26.84 | 14.56 | 50 | 27.64 | 13.83 | 50 |
| | IxG | 15.28 | 10.19 | 45.41 | 18.29 | 16.96 | 49.42 | 19.16 | 18.90 | 47.24 |
| | SIG | 27.02 | 3.40 | 64.77 | 29.34 | 14.05 | 59.09 | 37.31 | 5.10 | 66.38 |
| | DecompX | 52.16 | 0.92 | 84.48 | 57.94 | 2.41 | 63.27 | 44.86 | 2.72 | 66.76 |
| | ShapSampl | 43.96 | 0.46 | 82.87 | 58.29 | 2.44 | **71.29** | 44.86 | 2.43 | 67.17 |
| | LIME | 44.95 | 0.06 | 80.28 | 51.45 | 6.07 | 60.15 | 22.64 | 14.30 | 57.61 |
| Saloss BERT | Attention | 34.73 | 3.65 | 50 | 27.59 | 13.64 | 50 | 34.95 | 26.26 | 50 |
| | IxG | 14.98 | 12.66 | 51.01 | 24.19 | 16.30 | 49.02 | 26.61 | 30.94 | 50.74 |
| | SIG | 16.70 | 8.22 | 63.74 | 45.44 | 8.48 | 54.96 | 44.53 | 21.50 | 54.70 |
| | DecompX | 59.37 | 0.30 | 75.34 | 59.42 | 5.38 | 62.02 | 58.71 | 13.23 | 65.17 |
| | ShapSampl | 37.73 | 0.77 | 73.96 | 65.38 | 3.17 | 70.23 | 57.05 | 15.10 | 72.36 |
| | LIME | 53.18 | 2.37 | 76.16 | 53.31 | 6.32 | 58.65 | 21.73 | 21.96 | 55.71 |
| B-cos BERT | B-cos | **64.22** | **-1.26** | **87.92** | **74.18** | **-2.87** | 70.43 | **59.66** | **-4.89** | **77.57** |

Table 2: Faithfulness evaluation for conventionally fine-tuned, Saloss, and B-cos BERT across three datasets. The best results are in **bold**. B-cos explanations are consistently more faithful than post-hoc explanations from both baseline models.

## 4.3 Human Evaluation

Contrary to previous B-cos studies that rely solely on automatic evaluations to assess explanations, we conduct the first human study to better evaluate the human interpretability and agreement of B-cos explanations. We compare them against three strong post-hoc explanation methods on conventional BERT models.

Following the practice in Enguehard (2023) and Yue et al. (2022), we randomly select 50 instances, respectively, from AG News and HateXplain where the B-cos and conventional models make the same prediction. Five annotators then rate the explanations in terms of human interpretability (how well they understand them) and human agreement (how much they agree with them) on a scale of 1-5. Further details on the evaluation criteria, rating scales, annotator instructions, and example annotations can be found in Appendix J.

Figure 3 shows that B-cos explanations have a better human interpretability and exhibit greater alignment with human reasoning than post-hoc methods, even though they are not directly optimized for human agreement. Paired t-tests with a Bonferroni-corrected significance level $\alpha = \frac{0.05}{6} = 0.008\overline{3}$ (Bonferroni, 1936) shows that the improvements of B-cos explanations are statistically significant ($p < \alpha$) for both metrics.

## 4.4 Qualitative Analysis

Figure 4 provides an example of B-cos and other (post-hoc) explanations. It can be seen that the B-cos explanation highlights important tokens correctly with little focus on irrelevant ones. In contrast, ShapSampl attributes the

Figure 3: Human evaluation reveals that B-cos explanations have better human interpretability and human agreement than baseline methods.

highest importance to the [SEP] token and provides only little useful information. Meanwhile, DecompX extracts a significant amount of irrelevant information. Overall, the B-cos explanation is more interpretable to humans by providing clearer and more relevant attributions compared to the post-hoc explanations. We provide more examples and analyze two undesired explanations from B-cos LMs in Appendix K.

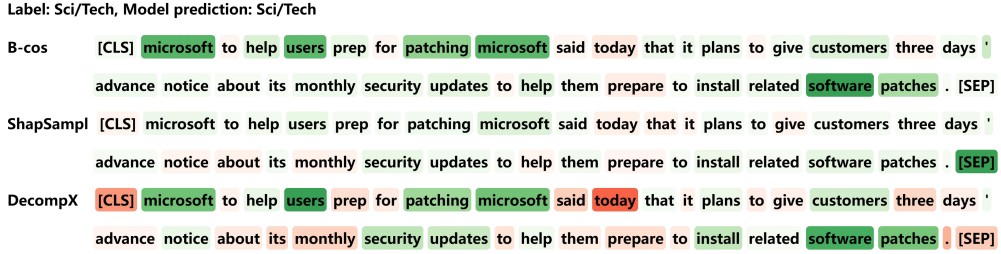

Figure 4: Examples of B-cos explanations (B-cos BERT) as well as ShapSampl and DecompX explanations (conv. BERT) from AG News. Green (red) indicates the positive (negative) impact of tokens on the prediction. The B-cos explanation highlights only relevant tokens and is more interpretable to humans.

## 4.5 Ablation Study

To better understand B-cos LMs, we conduct an ablation study evaluating the impact of key design choices on task performance and explanation faithfulness.

|  | Acc ($\uparrow$) | Comp ($\uparrow$) | Suff ($\downarrow$) | SeqPG ($\uparrow$) |
|---|---|---|---|---|
| Full system | 78.64 | 59.66 | -4.89 | 77.57 |
| w/o alignment pressure (B=1) | 78.07 (0.57) | 57.19 (2.44) | -2.57 (2.32) | 70.18 (7.39) |
| w/o BCE training | 79.00 (0.36) | 49.22 (10.44) | -7.91 (3.02) | 79.21 (1.64) |
| w/o architectural adaptations | 77.65 (0.99) | 52.23 (7.43) | -3.80 (1.09) | 74.30 (3.27) |
| w/o dynamic linear weights (IxG) | 78.64 (0.00) | 44.93 (14.73) | -0.60 (4.29) | 53.57 (24.00) |
| $\mathbf{W(x)x}$ from conv. model | 80.77 (2.13) | 44.92 (14.74) | 2.80 (7.69) | 70.20 (7.37) |

Table 3: Ablation study of key designs in the B-cos BERT model on HateXplain. Green (red) values in parentheses indicate the results are better (worse) than the full system.

In Table 3, we find that removing alignment pressure (using B=1) degrades both task performance and explanation faithfulness. Replacing cross-entropy with BCE loss has little effect on classification accuracy,

but improves faithfulness in perturbation-based evaluations. Architectural adaptations, including removing bias terms and eliminating activation functions in prediction heads, are also critical for enhancing task performance and explainability. Besides, we observe numerical instability when generating explanations without these architectural adaptations, as the dynamic linear weights for sigmoid and tanh ($\text{sigmoid}(\mathbf{x}) \times \mathbf{x^{-1}}$ and $\tanh(\mathbf{x}) \times \mathbf{x^{-1}}$) become unstable when $\mathbf{x}$ is close to zero.

In addition to ablations of model design and training components, we also evaluate alternative explanation methods. Replacing the dynamic linear weights $\mathbf{W}(\mathbf{x})$ with gradients (equivalent to IxG) yields less faithful explanations on B-cos LMs. Besides, directly extracting B-cos-like explanations, $\mathbf{W}(\mathbf{x})\mathbf{x}$, from a conventional model results in worse faithfulness compared to those from B-cos LMs.

## 5 Comparison of B-cosification Setups

Transforming PLMs into task-specific B-cos LMs involves two key objectives: task fine-tuning and B-cos conversion. While our main experiments combine these two phases, they can also be performed separately. To assess their effects, we compare two alternative training setups:

- Task then B-cos: PLMs are first fine-tuned on a downstream task. B-cos adaptations are then applied, followed by further fine-tuning on the same task for B-cos conversion. This setup is equivalent to Arya et al. (2024) who apply B-cosification to models with downstream task capabilities.
- B-cos then task: B-cos adaptations are applied to PLMs first, followed by unsupervised pre-training to enhance B-cosification. The pre-trained B-cos models are then fine-tuned on the downstream task.

We evaluate these setups against the B-cosification approach used in our main experiments (B-cos LM) and compare task performance, faithfulness, and training efficiency (cf. Appendix D for B-cos pre-training details). We also report results for conventional fine-tuning (Conv. LM) and randomly initialized B-cos models (B-cos from scratch). All experiments are run on IMDB with B=1.25 for B-cos models, with results averaged over three runs.

Table 4 shows that B-cos LM requires fewer training steps to reach optimal validation performance than conventional fine-tuning. Training B-cos LM from scratch results in worse accuracy and faithfulness, showing the importance of good parameter initialization. Among the two setups that separate task fine-tuning and B-cos conversion, *Task then B-cos* achieves results similar to B-cos LM but requires more total training steps. *B-cos then task* initially performs worse under the same training budget. However, with additional pre-training epochs, it surpasses other B-cosification setups in both task performance and faithfulness. Overall, we find that combining task fine-tuning and B-cos conversion is the most efficient approach.

| Setup | Epochs | Acc (↑) | SeqPG (↑) | Steps (K) |
|---|---|---|---|---|
| Conv. LM | 5 | 94.06 | - | 6.67 |
| B-cos LM | 5 | 91.00 | 70.66 | 4.33 |
| B-cos from scratch | 5 | 88.25 | 60.92 | 4.33 |
| Task then B-cos | 1+4 | 91.17 | 70.01 | 1+5 |
| | 2+3 | 91.30 | 70.48 | 3+3.33 |
| | 3+2 | 91.38 | 70.83 | 4+3 |
| | 4+1 | 89.56 | 70.66 | 5+1 |
| | 5+5* | 91.27 | 70.78 | 6.67+3.33 |
| B-cos then task | 1+4 | 90.64 | 67.07 | 1+5 |
| | 2+3 | 91.04 | 68.97 | 3+4 |
| | 3+2 | 90.50 | 68.48 | 4+3 |
| | 4+1 | 89.18 | 69.92 | 6+1 |
| | 5+5* | 91.45 | 71.86 | 7+5.33 |
| | 10+5* | 92.19 | 73.44 | 15+6.33 |
| | 20+5* | 92.87 | 75.01 | 31+6 |

Table 4: Training epochs, accuracy, explanation faithfulness, and convergence steps for different B-cosification setups. For two-phase methods, we report epoch distribution and convergence steps per phase. * marks additional training epochs.

However, with sufficient pre-training, *B-cos then task* can produce more performant and explainable models.

## 6 Impact of B-cosification and B Values

For a deeper understanding of how B-cosification and alignment pressure parameter B affect model performance and behavior, we compare conventional and B-cos BERT trained on HateXplain across different B values. We also provide an empirical analysis of the impact of B on input-weight alignment in Appendix L.

**Model Performance** Figure 5 shows the effects of varying B on the task performance and explanation faithfulness. Classification accuracy initially improves slightly as B increases from 1 to 1.25, benefiting from

the extra non-linearity introduced by B>1. However, beyond this point, accuracy declines as higher alignment pressure reduces model flexibility. A similar trend is observed for Comp, which peaks around B=1.5 before decreasing. This differs from previous findings in vision models (Böhle et al., 2022), which we attribute to the high sparsity of explanations at larger B values. As alignment pressure increases, fewer tokens receive attribution scores that are not close to zero, leading to poor token importance calibration and worse Comp scores. The effects of B on other metrics are similar and can be found in Appendix M.

**Explanation Entropy**   Figure 5 also reveals a negative correlation between explanation entropy and B, indicating that higher alignment pressure leads to sparser explanations. This aligns with our expectations: a larger B amplifies the differences between dimensions in $|\cos(\mathbf{x}, \hat{\mathbf{W}})|^{B-1}$ of B-cos layers (Equation 2) and the dynamic linear weight assigns more distinct attributions to input features. As a result, explanations become more concentrated, where only a few tokens receive high attributions, while most remain close to zero (cf. Appendix N for an example).

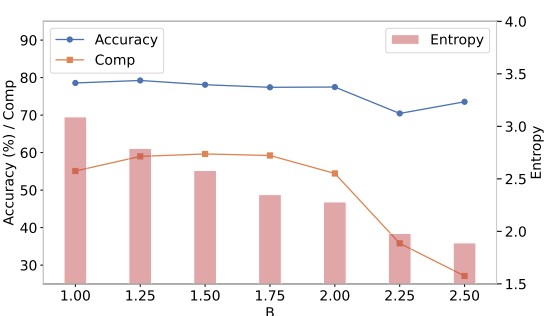

Figure 5: Varying B for B-cos BERT (HateXplain). Accuracy and Comp both peak around B=1.5, while explanation entropy negatively correlates with B.

**Model Bias**   PLMs often acquire biased patterns present in their training data (Wang & Demberg, 2024a;b). Since B-cos LMs with larger B values rely on fewer tokens for prediction, we investigate whether this may cause them to overfit and learn biases in the data. For this, we examine label bias and word-level spurious correlations using HateXplain, where approximately 60% of training and test examples have positive labels and societal biases are present. Figure 6 shows that a larger B value (B=2.5) reduces the model capacity, leading to a substantially higher positive rate in predictions and therefore lower class-balanced accuracy. Moreover, the B=2.5 model assigns higher attributions to non-semantic [CLS] and [SEP] tokens, indicating a reduced reliance on meaningful content. Notably, this label bias is not observed in the balanced AG News and IMDB datasets.

We also find that B-cosification, particularly with large B, amplifies reliance on spurious correlations. For example, the prediction positive rate for examples with the word "black" rises from 49.02% in the test set and 52.94% in the conventional model to 59.80%, 56.86%, and 73.53% in B-cos LMs with B=1, 1.5, and, 2.5, respectively (we provide an example in Appendix O). However, the faithfulness and interpretability of B-cos explanations facilitate the detection of spurious correlations and can effectively guide models toward reducing them (Rao et al., 2023; Balkir et al., 2022; Wang et al., 2025b). We leave the exploration of B-cos LMs for bias detection and mitigation to future work.

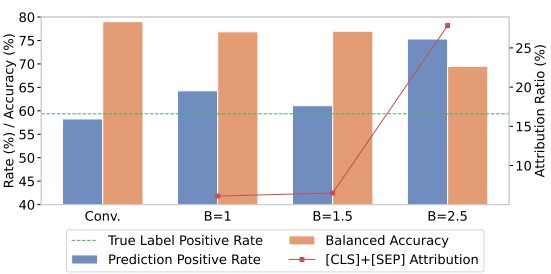

Figure 6: Comparison of conv. BERT and B-cos BERT with different B values. The attributions to [CLS] and [SEP] tokens (■) indicate that B-cos LMs with large B overfit to the non-semantic label distribution.

# 7   B-cosifying Decoder-Only Models for Generation Tasks

LLMs are increasingly used as general-purpose assistants, with most based on decoder-only architectures (Zhao et al., 2023; Minaee et al., 2024). While our primary focus is on classification tasks using encoder-only models, we also extend B-cosification to decoder-only models for generation tasks to demonstrate the broader applicability of B-cos LMs. Specifically, we apply B-cosification to two decoder-only models, GPT-2 small (Radford et al., 2019, referred to as GPT-2 afterwards) and Llama-3.2-1B (Dubey et al., 2024, referred to as Llama-3.2 afterwards), and evaluate their language modeling performance and explanation quality on two generation tasks. For more details on the datasets, experimental setup and baseline models, see Appendix D.

**B-cosification Setup** Given the complexity of modeling natural language, we use a small B value of 1.1. We do not B-cosify the language head, as its parameters are tied with the embedding layer. We use the standard cross-entropy loss instead of BCE, since the unnormalized language head weights could otherwise grow arbitrarily large to minimize the loss. To convert GPT-2 and Llama-3.2 to B-cos LMs, we apply B-cos adaptations and further train them on 500,000 and 4,000,000 sentences from Open-WebText[3], respectively.

**Datasets** For explanation evaluation, we use the BLiMP dataset (Warstadt et al., 2020) to assess explanations for linguistic phenomena, and the Indirect Object Identification (IOI) dataset (Brian Muhia, 2022) to test models' reasoning about object identification.

| Model | Probability Gap (↑) | | PPL (↓) |
|---|---|---|---|
| | BLiMP | IOI | |
| GPT-2 | 0.0055 | 0.3351 | 3.10 |
| B-cos GPT-2 | **0.0059** | 0.3265 | **3.04** |
| Llama-3.2 | 0.0058 | 0.4652 | 2.51 |
| B-cos Llama-3.2 | **0.0065** | **0.5021** | 2.64 |

Table 5: Language ability results for vanilla and B-cos decoder-only models. Scores where B-cos LM outperforms their vanilla counterparts are in **bold**. B-cos LMs show language modeling ability comparable to vanilla models. Results for each subset can be found in Table 13 in Appendix P.

Following Ferrando et al. (2023), we use nine subsets of BLiMP. Each example in both datasets consists of a sentence prefix followed by a target and a foil next word prediction, differing in whether they align with the phenomenon or ability of interest. Ground truth evidence is provided to support either grammatical correctness or correct object identification. Examples of these datasets can be found in Table 7 in Appendix D.

**Metrics and Baselines** We evaluate explanation quality using Mean Reciprocal Rank (MRR), where higher scores indicate stronger alignment with the ground truth evidence. To assess language modeling abilities of models, we report two metrics: (1) the probability gap between target and foil predictions, and (2) perplexity (PPL) on a held-out corpus. Following Yin & Neubig (2022), we generate contrastive explanations that explain why the model predicts target tokens instead of foil tokens, and compare B-cos explanations against several baseline methods: L1 gradient norm (Grad Norm), IxG, Occlusion, and two propagation-based methods Logit and ALTI Logit from Ferrando et al. (2023).

| Method | GPT-2 MRR (↑) | | Llama-3.2 MRR (↑) | |
|---|---|---|---|---|
| | BLiMP | IOI | BLiMP | IOI |
| Random | 0.5130 | 0.2360 | 0.5132 | 0.2328 |
| Grad Norm | 0.5465 | 0.8599 | 0.5504 | 0.3637 |
| IxG | 0.4750 | 0.1112 | 0.5303 | 0.1034 |
| Occlusion | 0.6365 | 0.8517 | 0.6201 | 0.4767 |
| Logit | 0.7307 | **1.0** | - | - |
| ALTI Logit | 0.7391 | **1.0** | - | - |
| B-cos | **0.7561** | 1.0 | **0.6969** | **0.9913** |

Table 6: Alignment results (MRR) on BLiMP and IOI. Logit and ALTI Logit results are replicated from the original paper (Ferrando et al., 2023). Best scores are marked in **bold**. B-cos explanations achieve the best alignment with ground truth evidence. Results for each subset can be found in Table 14 and Table 15 in Appendix P.

**Results** Table 5 shows that B-cos GPT-2 and B-cos Llama-3.2 models achieve strong language modeling performance comparable to their vanilla counterparts. Besides, Table 6 demonstrates that B-cos explanations exhibit better alignment with ground truth across tasks and models, indicating improved explainability of B-cos decoder-only LMs. Although the current B-cosification pipeline requires additional training, future work could explore more efficient approaches that reduce training overhead or integrate B-cosification into the pre-training phase. Overall, we believe B-cos decoder-only models are well-suited for tasks where explainability is critical and represent a promising direction for building more transparent and reliable LLMs.

## 8 Practical Guidance for Training B-cos LMs

Based on our experiments and analyses, we provide the following guidance on configuring B-cos LMs:

**B-cosification Setup** For efficient B-cosification of encoder-only models, we recommend combining B-cos conversion and task fine-tuning. However, if resources permit, an additional B-cos pre-training stage can further enhance both task performance and explanation faithfulness. For decoder-only models, B-cosification

---

[3]https://huggingface.co/datasets/Skylion007/openwebtext

can be applied to pre-trained models with unsupervised training data, enabling their conversion into B-cos LMs with improved explainability.

**Choice of B Value**   Several factors influence the selection of an appropriate B value:

(1) Model capacity and explanation sparsity: Excessively large B values can reduce model capacity and produce overly sparse explanations. We therefore recommend setting B within the range of 1–2.

(2) Task complexity and language diversity: For complex tasks with varied language use (e.g., online forum data with diverse language styles), smaller B values are preferable, as they preserve model flexibility and capture more useful features. For other tasks, larger B values can improve explainability.

(3) Bias considerations: Be mindful that B-cosification may amplify biases that the model learns from biased training data. When this risk is present, consider choosing smaller B values and applying bias mitigation methods if necessary.

## 9   Conclusion

In this work, we introduce B-cos LM, a bias-free dynamic linear model that learns task-relevant patterns through increased input-weight alignment pressure. B-cos LMs generate more faithful and human interpretable explanations while maintaining strong task performance and fast convergence. We further explore adapting decoder-only models into B-cos LMs for generation tasks and show that, with additional training, they match the language modeling performance of conventional models while providing better explanations. Finally, based on our systematic analysis, we provide practical guidelines for effectively transforming PLMs into B-cos LMs.

## 10   Limitations

This study has certain limitations that should be acknowledged. First, the automatic evaluation metrics we use may not fully capture the faithfulness of different explanation methods (Feng et al., 2018; Lapuschkin et al., 2019). However, since there is no universal consensus on the most reliable evaluation metrics, this remains an open challenge in explainability research.

Second, we find that B-cos explanations do not consistently capture token interactions within multi-token phrases. For example, a negation phrase like *not good* tends to receive an overall attribution score that aligns with its meaning (e.g., a negative score for positive sentiment), but the individual token scores within the phrase vary across contexts. In some cases, the word *good* may receive either positive or negative scores across different examples, even when the overall sentiment remains the same. Similar issues arise in other methods, suggesting a broader limitation of token-level rationales in capturing compositional semantics.

Finally, B-cos explanations, like all input-based approaches, are not suitable for all tasks. They are particularly suitable for NLP tasks where predictions can be directly attributed to specific words or phrases. Examples include various text classification tasks, such as sentiment analysis and toxicity detection, where the presence or absence of certain tokens often provides clear evidence for the model's decision. Conversely, for tasks requiring multi-step reasoning or external knowledge (e.g., commonsense reasoning, fact verification), token-level rationales become less informative. Higher-level explanations, such as concept-based or natural language explanations, may instead offer more meaningful insight into the model's reasoning.

## Acknowledgements

This work was funded in part by the Deutsche Forschungsgemeinschaft (DFG, German Research Foundation) – GRK 2853/1 "Neuroexplicit Models of Language, Vision, and Action" - project number 471607914. We also thank all those who provided valuable feedback and suggestions during the development of this work. Their input has been instrumental in improving its quality.

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

## A Terminology

To ensure clarity, we define key terms used in this work as follows:

- **Faithfulness** The extent to which an explanation accurately reflects the model's actual reasoning process (Jacovi & Goldberg, 2020). A faithful explanation should directly correspond to the internal mechanisms that led to the model's prediction.
- **Human Interpretability** The ease with which a person can understand the model's reasoning from the explanation (Lage et al., 2019). A highly interpretable explanation should be clear, concise, and focused on relevant information while avoiding unnecessary or distracting information. However, an explanation that is easy for humans to interpret may not necessarily reflect the model's actual reasoning process or align with human reasoning patterns.
- **Human Agreement** The degree to which a model's explanation aligns with the reasoning a human would use for the same prediction. A high-agreement explanation should follow intuitive, logical reasoning patterns similar to human decision-making.
- **Explainability** The extent to which a model's computations can be faithfully explained and its learned patterns are understandable to humans. A highly explainable model should yield explanations that are both faithful to its actual reasoning process and interpretable to humans.

## B Notation

In this paper, we use lowercase letters for scalars (e.g., b), bold lowercase letters for vectors (e.g., $\mathbf{w}$, $\mathbf{x}$), and bold uppercase letters ($\mathbf{W}$) for matrices. A special case is the alignment pressure parameter, denoted by the non-bold uppercase letter B, to distinguish it from the bias term b in linear layers. We use bold uppercase letters $\mathbf{X}$ and $\mathbf{A}$ to denote a sequence of model inputs or hidden state activations. In § 3, we use $\mathbf{x}$ to denote the input when a function is applied to each element of the input sequence separately. In contrast, we use $\mathbf{X}$ or $\mathbf{A}$ when the function involves interactions between elements, such as in the attention mechanism.

## C Dynamic Linear Representation of Model Components

Here we describe how each model component in transformers can function as or be converted to a bias-free, dynamic linear module in B-cos LMs.

**B-cos Layers** B-cos layers are designed as bias-free, dynamic linear modules with a dynamic linear weight matrix $\mathbf{W}(\mathbf{x}) = |\cos(\mathbf{x}, \hat{\mathbf{W}})|^{\text{B-1}} \otimes \hat{\mathbf{W}}$. Here, $\otimes$ scales the rows of the matrix $\hat{\mathbf{W}}$ to its right by the scalar entries of the vector to its left.

**Non-linear Activation Functions**   In transformer models, non-linearity is typically introduced using (approximately) piecewise linear activation functions, such as ReLU (Nair & Hinton, 2010) and GELU (Hendrycks & Gimpel, 2016). These functions can be easily interpreted as linear transformations with input-dependent weights. For example, $\text{GELU}(\mathbf{x}) = \mathbf{x} \times (0.5 + 0.5 \times \text{erf}(\mathbf{x}/\sqrt{2}))$ can be interpreted as a linear transformation where the second term acts as the dynamic linear weight.

**Attention Blocks**   Böhle et al. (2024) showed that attention computations can be seamlessly integrated into B-cos networks as a dynamic linear module:

$$\text{Att}(\mathbf{X}; \mathbf{Q}, \mathbf{K}, \mathbf{V}) = \text{softmax}(\mathbf{X^T Q^T K X})\mathbf{V X} = \mathbf{A(X)VX} = \mathbf{W(X)X} \tag{4}$$

For multi-head self-attention (MSA), the output can be viewed as the concatenation of the outputs from $H$ attention heads, followed by a linear projection with matrix $\mathbf{U}$:

$$\text{MSA}(\mathbf{X}) = \mathbf{U}[\mathbf{W}_1(\mathbf{X})\mathbf{X}, ..., \mathbf{W}_H(\mathbf{X})\mathbf{X}] \tag{5}$$

Since this operation maintains a dynamic linear structure, the multi-head attention block remains a dynamic linear module.

**Normalization Layers**   Following Böhle et al. (2024), we apply bias-free normalization layers to ensure completeness of explanations:

$$\star\mathbf{Norm}(\mathbf{x}, \mathcal{X}; \gamma) = \frac{\mathbf{x} - \langle \mathcal{X} \rangle_\star}{\sqrt{\text{var}_\star(\mathcal{X})}} \times \gamma \tag{6}$$

where $\mathcal{X}$ represents a batch or sequence of inputs and $\star$ is the dimension along which the mean $\langle \cdot \rangle$ and variance $\text{var}(\cdot)$ are computed (e.g., across the batch or layer). Unlike standard normalization, this variant omits the bias term in the affine transformation to preserve explanation completeness. If a running mean estimate is used during inference, the centering term $\langle \mathcal{X} \rangle_\star$ is also removed. This yields a bias-free, dynamic linear transformation with weight $\sqrt{\text{var}_\star^{-1}(\mathcal{X})} \times \gamma$.

## D   Implementation Details

**Fine-tuning Setups**   For all PLMs used in the experiments, we use the uncased base version from huggingface (Wolf et al., 2020). For both conventional models and B-cos LMs, we train them for 5 epochs with 10% linear warm-up steps on the downstream task datasets. The learning rates are set to 2e-5 for IMDB and HateXplain, and 3e-5 for AG News. All models use a batch size of 16 and a maximum sequence length of 512. For validation, we randomly sample half of the test set from IMDB and AG News.

**Baselines**   For IxG and ShapSampl, we use the Captum (Kokhlikyan et al., 2020) implementations.[4] We implement the Attention method ourselves, and LIME is sourced from the lit library[5]. For DecompX[6] and SIG[7], we use their official implementations with default configurations. The number of samples is set to 25 for ShapSampl and 3,000 for LIME, with [MASK] as the baseline token. For all explanation methods at the embedding level, model predictions are attributed to the combined sum of word, position, and token type embeddings (if applicable). In the main experiments, we compute token attribution scores by summing over all embedding dimensions, as this approach demonstrates better faithfulness results than using the L2 norm.

For Saloss models, we use the official codebase[8] with default hyperparameters to train BERT and RoBERTa on AG News, IMDB, and HateXplain. DistilBERT is not included, as it is not supported by the codebase.

In Section 7, we follow Ferrando et al. (2023) to generate contrastive explanations that highlight why the models predicts the target token instead of the foil token. For Occlusion explanations, we use the [PAD]

---

[4]https://captum.ai/api/

[5]https://github.com/PAIR-code/lit

[6]https://github.com/mohsenfayyaz/DecompX

[7]https://github.com/josephenguehard/time_interpret

[8]https://github.com/GChrysostomou/saloss

token to perform occlusion, instead of a zero vector as done in Yin & Neubig (2022) and Ferrando et al. (2023). Using zero vectors distorts the input distribution and, in generative settings, can influence predictions differently depending on position. To avoid such positional effects, we instead occlude using the in-distribution embedding of the [PAD] token.

**SeqPG Examples**   When constructing examples for SeqPG, we set the sequence length to 50 for AG News, 256 for IMDB, and 25 for HateXplain, aligning with their median lengths. Only examples longer than these thresholds are selected, and they are truncated to construct synthetic examples. Additionally, we only use examples that are correctly predicted with a minimum confidence of 75% after truncation. For a fair comparison, we evaluate Saloss models and B-cos LMs on the same sets of examples constructed based on the predictions of the corresponding conventional models.

**Automatic Evaluation Setups**   For task performance evaluation, we use the complete test set for each task. For faithfulness evaluation, we conduct perturbation-based evaluations on 2000 test examples and SeqPG on 500 test examples for AG News and IMDB. For HateXplain, we use the full test set for perturbation-based evaluation (1,924 examples) and construct 269, 310, and 308 SeqPG examples from it using BERT, DistilBERT, and RoBERTa, respectively. In the perturbation-based evaluation, the [CLS] token is never perturbed because it is used directly to make predictions.

**B-cos Pre-training**   For B-cos pre-training in § 5, we set B=1.25 and further pre-train the model on 25,000 sentences from the Wikipedia dataset[9] using masked language modeling loss with a learning rate of 1e-4 and a 15% masking ratio. We do not B-cosify the language head, as its parameters are tied with the embedding layer. Pre-training uses the standard cross-entropy loss rather than binary cross-entropy loss, since the unnormalized language head weights could otherwise grow arbitrarily large to minimize the loss.

**Decoder-only Models B-cosification**   We use the GPT-2 small and Llama-3.2-1B models from huggingface. As with the encoder-based models, we do not B-cosify the language head and use cross-entropy loss for GPT-2 and Llama-3.2 training. B-cos adaptations are first applied and both models are then trained on 500,000 and 4,000,000 sentences, respectively, from the OpenWebText dataset for one epoch, using a learning rate of 5e-4. For GPT-2, we use a batch size of 16 and a maximum sequence length of 512; for Llama-3.2, we use a batch size of 128 and a sequence length of 1024. Perplexity is evaluated on a held-out OpenWebText subset of 10,000 sentences using a maximum sequence length of 512.

**BLiMP Subsets**   We follow Ferrando et al. (2023) to use the following nine BLiMP subsets with corresponding IDs. aga: anaphor gender agreement; ana: anaphor number agreement; asp: animate subject passive; dna: determiner noun agreement 1; dnai: determiner noun agreement irregular 1; dnaa: determiner noun agreement with adj 1; dnaai: determiner noun agreement with adj irregular 1; npi: npi present 1; darn: distractor agreement relational noun. Examples of these datasets and the IOI dataset can be found in Table 7.

| Dataset | ID | Example |
|---|---|---|
| Anaphor gender agreement | aga | Katherine can't help herself / himself. |
| Anaphor number agreement | ana | Susan revealed herself / themselves. |
| Animate subject passive | asp | Amanda was respected by some waitresses / pictures. |
| Determiner noun agreement 1 | dna | Raymond is selling this sketch / sketches. |
| Determiner noun agreement irregular 1 | dnai | Adam hadn't discussed these analyses / analysis. |
| Determiner noun agreement with adjectives 1 | dnaa | Rebecca was criticizing those good documentaries / documentary. |
| Determiner noun agreement with adjectives irregular 1 | dnaai | Some waiters broke this lost foot / feet. |
| NPI present 1 | npi | Even Suzanne has really / ever joked around. |
| Distractor agreement relational noun | darn | A niece of most senators hasn't / haven't descended most slopes. |
| Indirect Object Identification | IOI | Friends Juana and Kristi found a mango at the bar. Kristi gave it to Juana / Kristi. |

Table 7: Examples from the BLiMP and IOI datasets. Green (red) indicates target (foil) predictions. Ground truth evidence for the correct continuations is underlined.

---

[9]https://huggingface.co/datasets/wikimedia/wikipedia

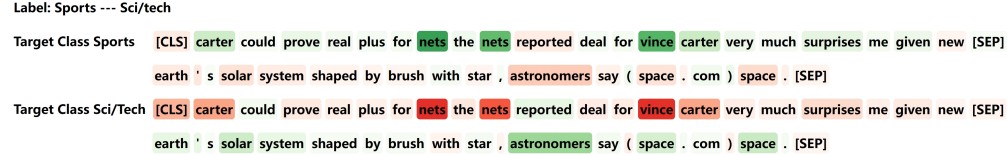

Figure 7: An example of SeqPG from AG News (using B-cos BERT). Green (red) indicates the positive (negative) impact of tokens on the prediction. The example consists of two sequences with different labels (Sports and Sci/tech), separated by the [SEP] token after the first sequence. Explanations are generated for each label, and the proportion of correctly attributed positive tokens is averaged across both labels to compute the SeqPG score for this example.

**Compute Infrastructure**  Unless stated otherwise, all experiments are conducted on a single NVIDIA H100 GPU. Training one epoch of B-cos BERT takes approximately 40 minutes on AG News, 10 minutes on IMDB, and 5 minutes on HateXplain.

# E  Explanation Efficiency

Beyond improved faithfulness and human interpretability, B-cos explanations are also efficient to extract. Comparing their computational costs with strong post-hoc methods shows that B-cos explanations are the most efficient in both time and memory usage (Table 8). Post-hoc and B-cos explanations are generated from the conventionally fine-tuned and B-cos BERT models on IMDB, respectively.

| Method | Time (s) | Memory (GB) |
|---|---|---|
| ShapSampl | 37.22 | 21.95 |
| LIME | 6.82 | 21.96 |
| SIG | 67.46 | 29.09 |
| DecompX | 0.76 | 48.38 |
| B-cos | **0.08** | **2.82** |

Table 8: Computational costs per example of generating explanations for 100 instances using an NVIDIA H100 GPU (batch size 1). B-cos explanations (**bold**) are at least 9x faster and require at most $\frac{1}{8}$ of VRAM.

# F  SeqPG Example

Figure 7 presents a SeqPG example from AG News using B-cos BERT. For better visualization, each segment is truncated to 20 tokens here instead of 50 used in the experiments. Unlike the hybrid document evaluation proposed by Poerner et al. (2018), our approach explicitly controls segment length and position to ensure a fair comparison. Additionally, we measure the proportion of correctly assigned positive attributions rather than relying solely on the highest attribution value.

# G  Task Performance of Other B-cos LMs

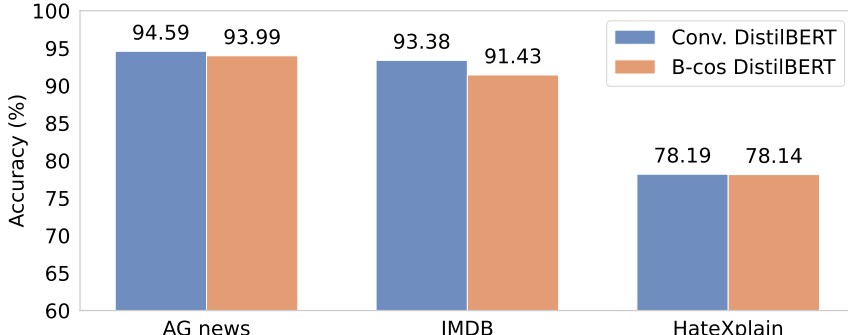

Figure 8: Mean accuracy of conventionally fine-tuned and B-cos DistilBERT models averaged over three runs. B-cos models perform comparably to conventional models on most tasks.

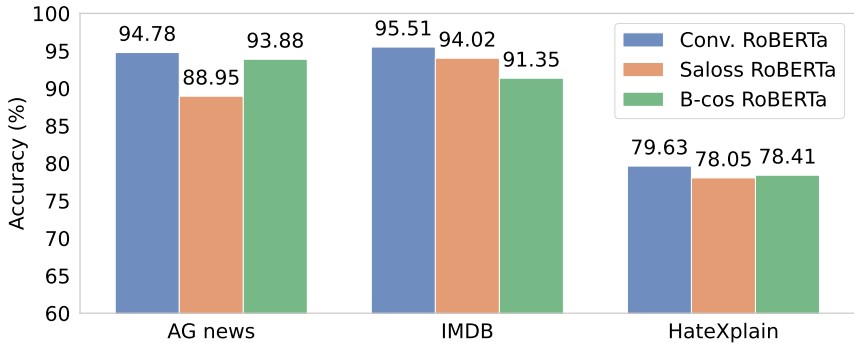

Figure 9: Mean accuracy of conventionally fine-tuned and B-cos RoBERTa models averaged over three runs. B-cos models perform comparably to conventional models on most tasks.

Figures 8 and 9 illustrate the task performance of conventional and B-cos DistilBERT and RoBERTa across datasets. Consistent with findings from BERT models (cf. Figure 2), B-cos LMs exhibit strong performance comparable to conventionally fine-tuned models.

## H   Faithfulness Evaluation of Other B-cos LMs

Tables 9 and 10 present the faithfulness evaluation results for DistilBERT and RoBERTa. The findings are consistent with our main experiments (cf. Table 2), confirming that B-cos LMs produce more faithful explanations compared to post-hoc explanation methods.

| Model | Method | AG News | | | IMDB | | | HateXplain | | |
|---|---|---|---|---|---|---|---|---|---|---|
| | | Comp ($\uparrow$) | Suff ($\downarrow$) | SeqPG ($\uparrow$) | Comp ($\uparrow$) | Suff ($\downarrow$) | SeqPG ($\uparrow$) | Comp ($\uparrow$) | Suff ($\downarrow$) | SeqPG ($\uparrow$) |
| Conv. DistilBERT | Attention | 26.36 | 5.37 | 50 | 31.62 | 10.46 | 50 | 30.56 | 14.67 | 50 |
| | IxG | 19.29 | 6.21 | 53.71 | 23.78 | 12.38 | 49.23 | 25.13 | 18.08 | 46.60 |
| | SIG | 30.78 | 1.63 | 67.87 | 47.16 | 5.48 | 60.66 | 41.11 | 4.23 | 58.55 |
| | DecompX | - | - | - | - | - | - | - | - | - |
| | ShapSampl | 52.56 | -0.56 | 82.64 | 63.29 | 2.91 | 70.27 | 48.73 | 0.87 | 64.44 |
| | LIME | 52.59 | -0.56 | 77.64 | 58.6 | 5.12 | 61.11 | 31.61 | 12.94 | 56.49 |
| B-cos DistilBERT | B-cos | **61.93** | **-1.01** | **86.78** | **76.26** | **-1.28** | **72.68** | **57.2** | **-4.49** | **74.89** |

Table 9: Faithfulness evaluation for conventionally fine-tuned DistilBERT and B-cos DistilBERT across three datasets. The best results are in **bold**. We find that B-cos explanations are consistently more faithful than post-hoc explanations from both models.

## I   Comparison to Rationale-Based Models

We compare B-cos LMs to one rationale-based, explain-then-predict BERT model, RGFS-SA (Saha et al., 2023)[10] on HateXplain. This model leverages human rationales as additional supervision during training. As shown in Table 11, although the RGFS-SA model brings improvement over the conventional BERT model, it generates considerably less faithful rationales compared to B-cos explanations.

## J   Human Evaluation Details

In the human study, we select only examples shorter than 25 tokens for HateXplain and 40 tokens for AG News to improve visualization. Additionally, we replace [CLS] and [SEP] with ## to make the examples more understandable for lay users. Below, we provide the instructions along with a detailed description of the criteria and scoring used in our human evaluation. In our human study, 92% of AG News examples and 80%

---

[10]https://huggingface.co/Hate-speech-CNERG/Rationale_predictor

| Model | Method | AG News | | | IMDB | | | HateXplain | | |
|---|---|---|---|---|---|---|---|---|---|---|
| | | Comp (↑) | Suff (↓) | SeqPG (↑) | Comp (↑) | Suff (↓) | SeqPG (↑) | Comp (↑) | Suff (↓) | SeqPG (↑) |
| Conv. RoBERTa | Attention | 22.17 | 3.80 | 50 | 25.26 | 5.84 | 50 | 32.94 | 7.52 | 50 |
| | IxG | 11.33 | 7.54 | 44.15 | 16.15 | 11.53 | 47.20 | 24.40 | 15.16 | 50.59 |
| | SIG | 19.64 | 1.63 | 66.43 | 38.14 | 2.13 | 59.04 | 44.21 | -1.42 | 66.73 |
| | DecompX | 50.00 | -0.84 | **90.38** | 49.24 | 0.65 | 72.80 | 46.94 | -1.42 | 70.16 |
| | ShapSampl | 35.63 | -0.68 | 78.31 | 43.32 | 1.83 | 65.85 | 44.83 | -1.30 | 67.15 |
| | LIME | 19.28 | 2.85 | 66.73 | 21.07 | 8.32 | 50.81 | 27.97 | 11.38 | 58.59 |
| Saloss RoBERTa | Attention | 40.69 | 2.77 | 50 | 24.51 | 4.33 | 50 | 47.04 | 7.83 | 50 |
| | IxG | 6.19 | 27.30 | 52.46 | 10.98 | 12.30 | 47.92 | 22.78 | 25.78 | 49.49 |
| | SIG | 6.91 | 27.22 | 56.84 | 11.53 | 13.76 | 62.10 | 43.77 | 5.02 | 58.67 |
| | DecompX | 61.46 | 0.16 | 74.20 | 65.50 | 0.10 | **74.41** | 54.94 | 2.47 | 65.63 |
| | ShapSampl | 34.48 | 0.73 | 64.67 | 48.53 | 0.82 | 63.04 | **55.80** | 1.49 | 64.53 |
| | LIME | 15.93 | 8.03 | 55.17 | 18.04 | 6.47 | 50.94 | 29.62 | 15.78 | 56.00 |
| B-cos RoBERTa | B-cos | **62.47** | **-1.18** | 86.63 | **73.87** | **-2.30** | 74.05 | 51.33 | **-5.18** | **74.01** |

Table 10: Faithfulness evaluation for conventionally fine-tuned RoBERTa, Saloss RoBERTa and B-cos RoBERTa across three datasets. The best results are in **bold**. We find that B-cos explanations are consistently more faithful than post-hoc explanations from both models.

| Model | Method | Accuracy (↑) | Comp (↑) | Suff (↓) | SeqPG (↑) |
|---|---|---|---|---|---|
| Conv. BERT | Attention | 80.77 | 22.64 | 13.83 | 50 |
| RGFS-SA BERT | Rationale | 80.09 | 36.11 | 16.54 | 50 |
| B-cos BERT | B-cos | 78.64 | 59.66 | -4.89 | 77.57 |

Table 11: Performance of conventional, RGFS-SA and B-cos (B=1.5) BERT models on HateXplain. SeqPG is consistently 50 for rationale-based models, as their explanations are class-agnostic, similar to attention. The rationale-based RGFS-SA model generates less faithful explanations than B-cos BERT.

of HateXplain examples contain correct model predictions; in the remaining cases, explanations are supposed to support the wrong predictions.

**WARNING: SOME CONTENT IN THIS QUESTIONNAIRE IS HIGHLY OFFENSIVE.**

**Prerequisites:** Proficiency in English is required for this evaluation task. If you do not meet this criterion, please do not proceed.

We invite you to review 100 examples where LMs perform classification tasks and provide explanations for their predictions.

- The first 50 examples come from a hate speech detection task, where the model predicts whether a text is toxic or not toxic.
- The last 50 examples come from a topic classification task, where the model categorizes a text into one of four topics: sports, world, business, or sci/tech.

For each example:

- The model's prediction is shown along with four explanations justifying the prediction.
- The order of the explanations is randomized to prevent bias.
- Words highlighted in green indicate words that had a positive influence on the prediction, while words in red indicate words that had a negative influence. The intensity of the color reflects the strength of the impact.
- **Important:** The model's prediction may be incorrect. Your task is to evaluate the explanations based on how well they support the model's prediction, not the true labels.

Evaluation Task:

After reviewing each example, please rate the the **human interpretability** and **human agreement** of the four explanations on a scale of 1 to 5. Refer to the definitions and rating scales provided below when making your assessments.

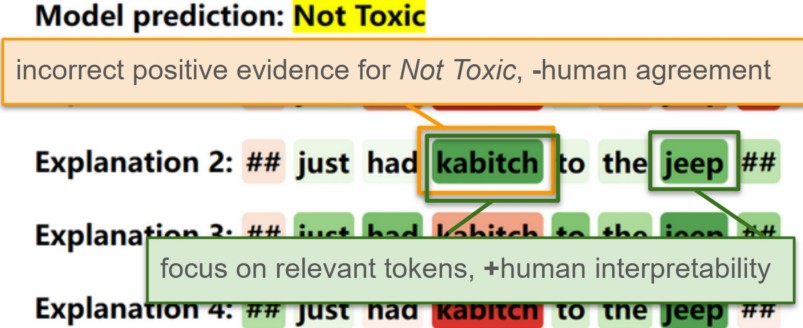

Figure 10: An example shown to participants that demonstrates how to rate explanations.

**Human Interpretability:** How easily a person can **understand the model's reasoning** based on the explanation. A highly interpretable explanation should be clear and easy to follow, focus on relevant words and avoid unnecessary or distracting details.

1. **Not Interpretable:** The explanation is unclear, noisy, or provides no meaningful insight.
2. **Slightly Interpretable:** Some clues are present, but the explanation is too sparse, irrelevant, or confusing.
3. **Moderately Interpretable:** The explanation contains useful information but is cluttered with noise or irrelevant details.
4. **Highly Interpretable:** The explanation is mostly clear, with minimal irrelevant highlights.
5. **Completely Interpretable:** The explanation is fully transparent, highlighting only the most relevant words, making the model's reasoning fully clear.

**Human Agreement:** How closely the model's explanation **aligns with the reasoning a human would use** for the same prediction. A high-agreement explanation should follow logical, intuitive reasoning and align with typical human decision-making patterns.

1. **No Agreement:** The explanation contradicts human reasoning or lacks logic.
2. **Low Agreement:** The explanation bears some resemblance to human reasoning but includes major inconsistencies.
3. **Moderate Agreement:** The explanation partially aligns with human reasoning, yet contains notable differences.
4. **High Agreement:** The explanation largely aligns with human reasoning, showing only minor discrepancies.
5. **Complete Agreement:** The explanation fully matches human reasoning, following a logical and intuitive path that a human would naturally use.

We also provide participants with examples to illustrate the reasoning behind rating explanations. One such example is shown in Figure 10. Additionally, Figure 11 presents an example of a model prediction and its explanations as displayed to participants during the study.

# K More Examples of B-cos Explanations

We provide two more examples of B-cos and other (post-hoc) explanations from AG News in Figure 12. Consistent with our findings in § 4.4, B-cos LMs provide more human interpretable explanations.

Figure 13 presents two examples where B-cos explanations do not align with human expectations. As B-cos explanations faithfully summarize model computations, they may also reveal spurious correlations from the

Words highlighted in **green** indicate words that had a **positive influence** on the prediction, while words in **red** indicate words that had a **negative influence**. The **intensity of the color** reflects the strength of the impact.

## symbols mark the beginning and end of the text.

Possible classes:

**Toxic**: The text contains language that is offensive, derogatory, or harmful toward individuals or groups, including insults, slurs, threats, or dehumanizing statements.

**Not Toxic**: The text does not contain harmful intent or offensive language, expressing opinions, criticism, or discussions in a respectful and non-threatening manner.

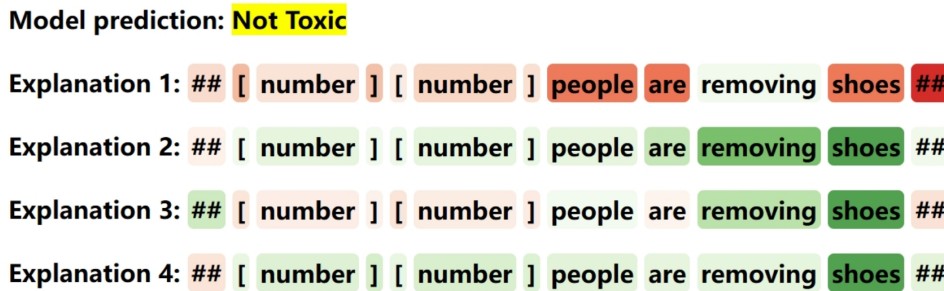

Figure 11: An examples of a model prediction and its explanations presented to participants.

training data and occasionally produce unreasonable rationales. For example, in the first case, the token "[user]" receives high attribution, likely due to recurring patterns in the data. Another observed error is polarity misinterpretation: in the second case, the token "niggas" is incorrectly highlighted as positive evidence for a non-toxic prediction. Similar types of undesired behavior have also been reported for other explanation methods.

## L  Impact of B on Input-weight Alignment

To analyze how B-cosification and alignment pressure influence the behavior of B-cos LMs, we compute the alignment (cosine similarity) between each input and its corresponding weight in B-cos modules across all layers. This analysis is performed on 100 examples from the HateXplain dataset. In Figure 14, we plot different percentiles of input-weight alignment for conventional and B-cos BERT models with varying B values. For better visualization, we display only the 10th to 90th percentiles.

Overall, larger B values generally lead to stronger input-weight alignment compared to smaller B and conventional models, as evidenced by the curves for B=1.5 and B=2.5 lying above those for the conventional model and B=1. However, the alignment pattern becomes more complex when comparing B=1.5 and B=2.5. Specifically, at B=2.5, the most aligned input-weight pairs exhibit higher alignment than in other models, but some pairs show very low alignment. This result may arise because certain weights are highly optimized for specific input patterns, leading to poor alignment with others, particularly in later layers where input features become more anisotropic (Ethayarajh, 2019; Li et al., 2020). As a result, some outputs from the B-cos layers are highly negative. When these outputs are fed into GELU activation functions, their dynamic weights approach zero, making the explanations more sparse.

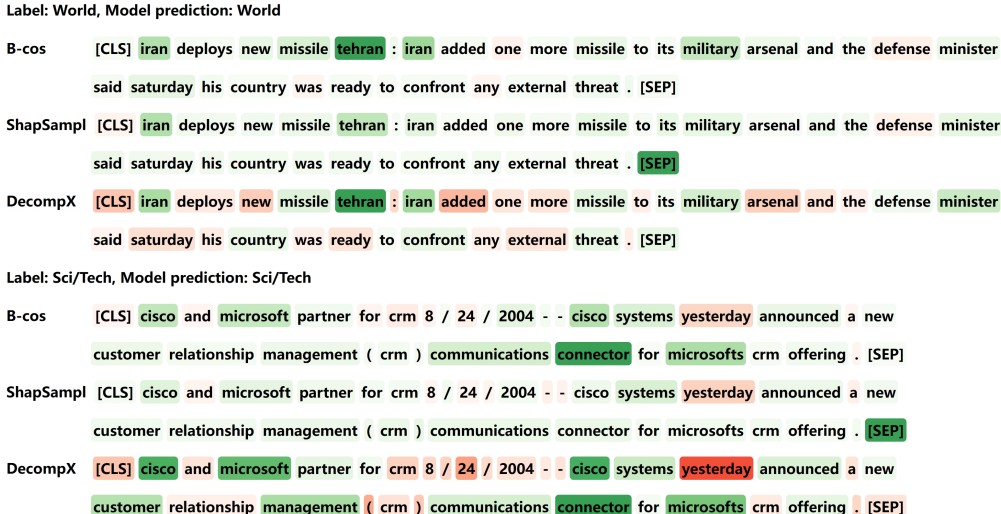

Figure 12: More examples of B-cos explanations (B-cos BERT) as well as ShapSampl and DecompX explanations (BERT) from the AG News dataset. Green (red) indicates the positive (negative) impact of tokens on the prediction. As can be seen, the B-cos explanation highlights only relevant tokens and is more interpretable to humans.

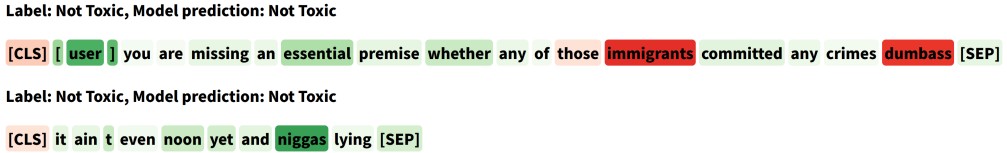

Figure 13: Two undesired B-cos explanations from B-cos BERT on HateXplain. Green (red) indicates the positive (negative) impact of tokens on the prediction. B-cos explanations can be influenced by spurious correlations learned by the model and may occasionally misattribute the contribution of certain tokens.

## M   Effects of B on Other Metrics

Table 12 presents the complete results on how B values affect task performance, explanation faithfulness and explanation entropy, as shown in Figure 5. Similar to Comp, SeqPG scores also decline with higher alignment pressure. This could also be attributed to the high sparsity of explanations. As B increases, fewer tokens receive attribution scores that are not close to zero, and in some SeqPG examples, B-cos LMs may attribute predictions to a single segment. This can lead to numerical instability when computing the positive attribution ratio.

| **B** | 1.00 | 1.25 | 1.50 | 1.75 | 2.00 | 2.25 | 2.50 |
|---|---|---|---|---|---|---|---|
| Acc (↑) | 78.57 | **79.23** | 78.10 | 77.41 | 77.48 | 70.44 | 73.55 |
| Comp (↑) | 55.09 | 58.99 | **59.64** | 59.23 | 54.44 | 35.80 | 27.11 |
| Suff (↓) | -4.25 | -5.71 | -5.47 | -5.84 | -6.69 | **-7.23** | -5.47 |
| SeqPG (↑) | 69.75 | 77.26 | **77.79** | 77.67 | 76.79 | 76.68 | 77.25 |
| Entropy | 3.09 | 2.79 | 2.58 | 2.35 | 2.28 | 1.98 | 1.89 |

Table 12: Task performance, explanation faithfulness, and explanation entropy of B-cos BERT models on HateXplain with different B values. Results are averaged over three runs. Similar to Figure 5, task performance and explanation faithfulness peak around B=1.5, while explanation entropy correlates negatively with B.

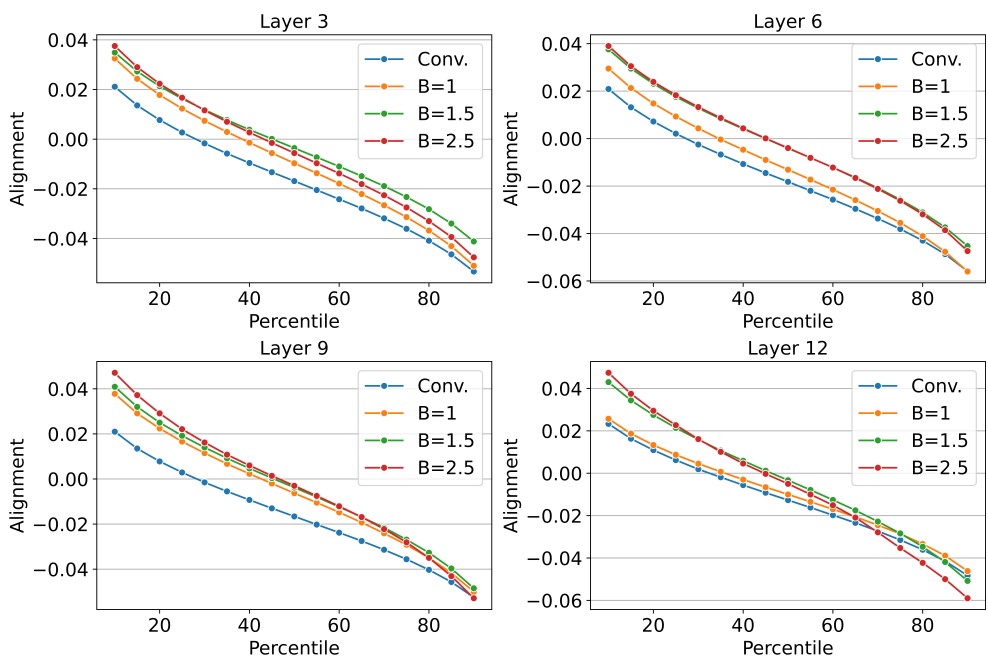

Figure 14: Percentiles of input-weight alignment in B-cos modules across selected layers of conventional and B-cos BERT models with different B values (HateXplain).

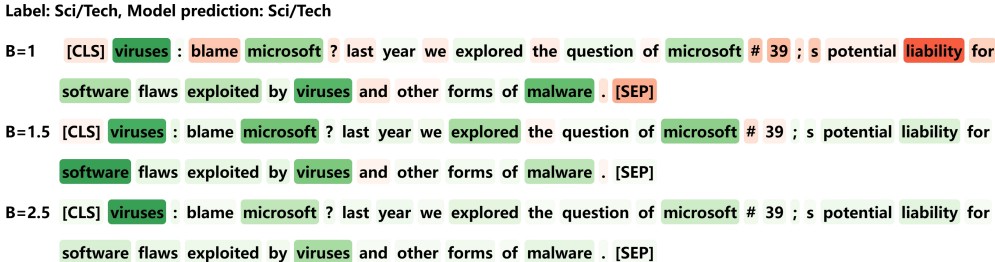

Figure 15: B-cos explanations (B-cos BERT) on AG News with different B values. Green (red) indicates the positive (negative) impact of tokens on the prediction. As B increases, B-cos LMs produce sparser explanations, with fewer tokens receiving significant attribution scores.

## N    B-cos Explanations with Different B Values

Figure 15 illustrates that with increased alignment pressure, B-cos LMs learn fewer but more task-relevant features. Consequently, they produce sparser explanations, with fewer tokens receiving significant attribution. This finding aligns with the statistics presented in § 6.

## O    Example of Model Bias

In the example shown in Figure 16, models become increasingly confident in the incorrect prediction as B increases, with attributions primarily assigned to the word "blacks". Moreover, simply replacing "blacks" with "whites" results in a sharp drop in confidence, which demonstrates a growing reliance on spurious correlations with increased alignment pressure. The observation further confirms our findings in §6.

**Label: Not Toxic, Model prediction: Toxic**

**B=1:   Prediction confidence=69.93, Confidence after perturbation=53.91**

[CLS] america is so racist that they think asians are smarter than blacks and hispanics ...

**B=1.5: Prediction confidence=92.54, Confidence after perturbation=8.93**

[CLS] america is so racist that they think asians are smarter than blacks and hispanics ...

**B=2.5: Prediction confidence=99.70, Confidence after perturbation=6.84**

[CLS] america is so racist that they think asians are smarter than blacks and hispanics ...

Figure 16: Example of how larger B values lead B-cos LMs to learn word-level spurious correlations. Green (red) indicates the positive (negative) impact of tokens on the prediction. Higher alignment pressure increases the reliance of B-cos LMs on spurious correlations in the data. In this example, perturbation involves changing "blacks" to "whites".

## P   Decoder-Only Model Results

Table 13 presents the average probability gaps between target and foil predictions on every dataset from different vanilla and B-cos models. Table 14 and Table 15 contain MRR results on every dataset in BLiMP and IOI for GPT-2 and Llama-3.2 models, respectively.

| Dataset | Probability Gap (↑) | | | |
|---|---|---|---|---|
| | **Vanilla GPT-2** | **B-cos GPT-2** | **Vanilla Llama-3.2** | **B-cos Llama-3.2** |
| aga | 0.0120 | 0.0170 | 0.0171 | 0.0196 |
| ana | 0.0152 | 0.0189 | 0.0140 | 0.0170 |
| asp | 0.0007 | 0.0008 | 0.0016 | 0.0009 |
| dna | 0.0011 | 0.0011 | 0.0011 | 0.0017 |
| dnai | 0.0021 | 0.0006 | 0.0011 | 0.0013 |
| dnaa | 0.0014 | 0.0012 | 0.0012 | 0.0017 |
| dnaai | 0.0091 | 0.0058 | 0.0077 | 0.0072 |
| npi | 0.0015 | 0.0002 | 0.0002 | 0.0001 |
| darn | 0.0067 | 0.0078 | 0.0080 | 0.0085 |
| IOI | 0.3351 | 0.3265 | 0.4652 | 0.5021 |

Table 13: Probability gaps between target and foil next token predictions from vanilla models and B-cos LMs on every dataset.

| Dataset | Random | Grad Norm | IxG | Occlusion | Logit | ALTI Logit | B-cos GPT-2 |
|---------|--------|-----------|-----|-----------|-------|------------|-------------|
| aga | 0.6875 | 0.7927 | 0.7910 | 0.7513 | 0.827 | 0.964 | 0.8764 |
| ana | 0.7056 | 0.6753 | 0.7387 | 0.5957 | 0.817 | 0.976 | 0.7532 |
| asp | 0.3818 | 0.7512 | 0.4086 | 0.4374 | 0.386 | 0.499 | 0.4939 |
| dna | 0.4608 | 0.3629 | 0.3869 | 0.9030 | 0.737 | 0.646 | 0.9308 |
| dnai | 0.4626 | 0.4077 | 0.4317 | 0.8395 | 0.711 | 0.637 | 0.8596 |
| dnaa | 0.4103 | 0.2632 | 0.3214 | 0.6557 | 0.951 | 0.807 | 0.7798 |
| dnaai | 0.4074 | 0.2632 | 0.3392 | 0.6167 | 0.9 | 0.757 | 0.7601 |
| npi | 0.6121 | 0.7854 | 0.4948 | 0.4775 | 0.445 | 0.417 | 0.4573 |
| darn | 0.4888 | 0.6170 | 0.3627 | 0.4247 | 0.802 | 0.949 | 0.8936 |
| IOI | 0.2360 | 0.8599 | 0.1112 | 0.8517 | 1.0 | 1.0 | 1.0 |

Table 14: MRR Alignment of different explanation methods on GPT-2 small predictions on every dataset. B-cos explanations are extracted from the B-cos GPT-2 model. Logit and ALTI Logit results are duplicated from Ferrando et al. (2023).

| Dataset | Random | Grad Norm | IxG | Occlusion | B-cos Llama-3.2 |
|---------|--------|-----------|-----|-----------|-----------------|
| aga | 0.6868 | 0.6030 | 0.5928 | 0.7811 | 0.8485 |
| ana | 0.7037 | 0.5955 | 0.6432 | 0.6072 | 0.8535 |
| asp | 0.3842 | 0.7694 | 0.4537 | 0.3670 | 0.6108 |
| dna | 0.4615 | 0.4598 | 0.4898 | 0.8352 | 0.6743 |
| dnai | 0.4630 | 0.4542 | 0.5043 | 0.7671 | 0.6652 |
| dnaa | 0.4112 | 0.4299 | 0.4750 | 0.6115 | 0.6308 |
| dnaai | 0.4075 | 0.4221 | 0.4498 | 0.5758 | 0.5563 |
| npi | 0.6123 | 0.6367 | 0.7062 | 0.5154 | 0.6264 |
| darn | 0.4884 | 0.5828 | 0.45787 | 0.5210 | 0.8065 |
| IOI | 0.2328 | 0.3637 | 0.1034 | 0.4767 | 0.9913 |

Table 15: MRR Alignment of different explanation methods on Llama-3.2 predictions on every dataset. B-cos explanations are extracted from the B-cos Llama-3.2 model. As Llama models are not supported in Ferrando et al. (2023), we do not include their results.

