# OpenReview forum: "B-cos LM: Efficiently Transforming Pre-trained Language Models for Improved Explainability"
_TMLR — Accepted by TMLR_

### Review · Reviewer_KuxQ · 2025-08-24

**Summary Of Contributions:**

Pre-trained language models achieve strong results in NLP tasks, but they are black-box systems. Post-hoc explanation methods (such as gradients, LIME, or SHAP) often generate unfaithful and hard-to-interpret explanations, making it difficult to trust their reasoning in sensitive domains. B-cos networks remove bias terms and enforce input-weight alignment helping improve explainability. The paper introduces B-cos LLMs, language models modified with the B-cos architecture. This makes explanations inherently faithful rather than post-hoc approximations. The paper presents and examines various strategies for transforming PLMs into task-specific B-cos LMs and shows that combining task fine-tuning and B-cos conversion is the most efficient approach.

**Additional Comments:**

No other comments.

**Audience:**

Yes

**Audience Explanation:**

The audience will be interested in this paper because it tackles one of the most pressing challenges: making powerful pre-trained language models explainable without sacrificing performance.

Current explanation methods are mostly post-hoc, often unfaithful to the model’s true reasoning, and difficult for humans to interpret. By introducing B-cos LMs, the authors provide a way to intrinsically embed explainability into the model architecture through bias-free transformations and input-weight alignment, while retaining nearly the same task performance. This approach is practical since it efficiently converts existing PLMs rather than requiring training from scratch, and timely, given increasing regulatory and societal demands for transparent AI. Moreover, the extension of B-cos LMs to decoder-only models shows promise for explainability in generative tasks, an area of growing importance as LLMs become central to real-world applications.

**Broader Impact Concerns:**

There are no broader impact concerns.

Beyond improving faithfulness and interpretability, B-cos explanations can also help uncover spurious correlations and biases, making them valuable not only for model trustworthiness but also for fairness and safety research, which are core concerns across the ML community. So, this work has positive broader impact.

**Claims And Evidence:**

Yes

**Claims Explanation:**

+ B-cos LM architecture for NLP that removes the bias term and used dynamic linear modules that yield faithful token-level attributions. The transformer components (attention, normalization, prediction heads) are adapted to bias-free B-cos modules. The linear laters are replaced with B-cos transformations that enforce cosine similarity alignment with input features. Fine-tuning with binary cross-entropy loss strengthens alignment. Explanations are derived directly from the learned input-weight alignment.

+ The paper shows that combining B-cos conversion and task fine-tuning yields faster convergence and better efficiency than prior methods.

+ Demonstrates superior explanation faithfulness and human interpretability across multiple datasets (AG News, IMDB (sentiment), HateXplain (hate speech)).  B-cos explanations outperform all post-hoc methods, with significant gains in faithfulness and human agreement. Ablation studies show that alignment pressure (B > 1) and architectural changes are crucial. Human studies confirm explanations are clearer and more aligned with human reasoning.

+ The extension to decoder-only models (e.g., GPT-2, Llama-3.2) for generation tasks, showing applicability beyond classification. Decoder-only B-cos models match language modeling ability of GPT-2 and Llama-3.2 while producing more faithful explanations.

**Requested Changes:**

The paper's presentation can be improved by reducing discussion covered by prior work on B-Cos networks and focusing on the paper's central contributions. For example, adding explanation about why B was set to different values for different datasets would be useful.

---

> ### Author Response · Authors · 2025-09-05
> **Reply to Reviewer KuxQ**
>
> We would like to thank the reviewer for their time and effort in providing insightful comments and suggestions. We have carefully considered each point raised and provided detailed responses. We hope that our clarifications and revisions address your concerns.
>
> # [Change 1]  Reducing B-cos network introduction & Adding guidance on B selection
>
> We greatly appreciate your helpful feedback. In the revised version, we will move Section 3.2 (Dynamic Linear Representation of Model Components) to the Appendix in order to streamline the main paper. In addition, we will add a dedicated section before the conclusion that discusses practical guidance on how to efficiently select the parameter B.
>
> ```
> Practical Guidance for Training B-cos LMs
>
> Based on our experiments and analyses, we provide the following guidance on configuring B-cos LMs:
>
> B-cosification Setup: For efficient B-cosification of encoder-only models, we recommend combining B-cos conversion and task fine-tuning. However, if resources permit, an additional B-cos pre-training stage can further enhance both task performance and explanation faithfulness. For decoder-only models, B-cosification can be applied to pre-trained models with unsupervised training data, enabling their conversion into B-cos LMs with improved explainability.
>
> Choice of B value: Several factors influence the selection of an appropriate B value:
> (1) Model capacity and explanation sparsity: Excessively large B values can reduce model capacity and produce overly sparse explanations. We therefore recommend setting B within the range of 1–2.
> (2) Task complexity and language diversity: For complex tasks with varied language use (e.g., online forum data with diverse language styles), smaller B values are preferable, as they preserve model flexibility and capture more useful features. For other tasks, larger B values can improve explainability.
> (3) Bias considerations: Be mindful that B-cosification may amplify biases that the model learns from biased training data. When this risk is present, consider choosing smaller B values and applying bias mitigation methods if necessary.
>
>
> ```

---

### Review · Reviewer_hr5c · 2025-08-25

**Summary Of Contributions:**

This paper extends the B-cos method to language models by removing bias terms to enhance model interpretability. The proposed approach first applies B-cos architectural modifications to a PLM and then fine-tunes it on downstream tasks. The resulting B-cos LM can generate explanations during inference, showing the contribution of inputs to decisions.
Strengths:1.Experiments were conducted on both encoder-only models for classification tasks and decoder-only models for generation tasks, demonstrating the method ’s versatility.2. Comprehensive experimental analysis was performed on applying the B-cos method to language models, highlighting the roles of different components.
Weaknesses:1. Compared to conventionally fine-tuned methods, B-cos shows lower accuracy . Isn ’t accuracy more important than interpretability, especially when the evaluation of interpretability is not yet fully reliable? 2.The classification tasks lack relatively complex test sets, such as arithmetic or reasoning tasks, and the same is true for generation tasks. It would be interesting to see results on more challenging datasets.

**Audience:**

Yes

**Audience Explanation:**

The method of applying B-cos to language models in this paper provides a more intuitive and clearer perspective for explaining the model ’s decisions, highlighting the contribution of inputs to the final prediction, and offering meaningful value in terms of model interpretability.

**Claims And Evidence:**

Yes

**Claims Explanation:**

Sections 4.2 and 4.3 experimentally verify the advantages of B-cos LM in terms of interpretability. Section 5 compares different conversion strategies, demonstrating the effectiveness of combining task fine-tuning with B-cos conversion. Section 7 validates the method ’s generality on generation tasks with decoder-only models.

**Requested Changes:**

1. It is important to know whether the method can achieve similarly good results on more challenging test sets. This approach can intuitively reveal how the model focuses on key information in complex tasks.
2. The consistent drop in accuracy compared to conventionally fine-tuned methods across different test sets needs a reasonable explanation . Although the faithfulness results are good, isn ’t accuracy more important?

---

> ### Author Response · Authors · 2025-09-05
> **Reply to Reviewer hr5c**
>
> We sincerely thank the reviewer for their valuable time and insightful feedback. We have carefully addressed each of the points raised and provided detailed responses. We hope that our clarifications and revisions adequately resolve your concerns.
>
> # [weakness 1] Results on more challenging tasks
>
> We agree that including more diverse tasks would make our findings more compelling. However, input-based explanations are not always suitable for tasks that require complex reasoning chains. In such cases, alternative approaches, such as natural language explanations, may provide more informative insights. For this reason, in the present work, we focus on tasks where predictions can be meaningfully attributed to input tokens. We consider extending B-cos explanations to more complex reasoning tasks as an important direction for future work. In the revised version, we will also include a more detailed discussion of the recommended use cases for B-cos explanations in the limitation section, clarifying where they are most suitable and where alternative forms of explanation may be preferable:
>
> ```
> Input-based explanations are not suitable for all tasks. They are particularly suitable for NLP tasks where predictions can be directly attributed to specific words or phrases in the input. Examples include various text classification tasks, such as sentiment analysis and toxicity detection, where the presence or absence of certain tokens often provides clear evidence for the model’s decision. In such settings, token-level explanations are both intuitive and precise, enabling users to easily verify whether the model attends to the expected parts of the input. On the contrary, in tasks that involve multi-step reasoning or external knowledge retrieval such as commonsense reasoning or fact verification, token-level explanations are often less informative. In these cases, higher-level explanations, such as concept-based or natural language explanations, may better capture the underlying reasoning process and provide users with more meaningful insights.
> ```
>
>
> # [weakness 2] Loss in task performance
>
> The observed drop in task performance is mainly due to reduced model flexibility caused by stronger alignment pressure and the removal of bias terms, as shown in our ablation studies and analyses (Sections 4.5 and 6). While we agree that task performance is important, faithful and human-interpretable explanations are also crucial, and in some applications mandatory (see Section 1). Given that our method yields substantial improvements in explainability (Table 2, Figure 3) with only marginal reductions in accuracy (Figure 2), we believe B-cos LMs strike a practical balance and remain useful for many applications.

---

> > ### Comment · Reviewer_hr5c · 2025-09-11
> >
> > Thank you for the clarification. For the first weakness, while I understand that input-based explanations may be less effective beyond text classification, I would encourage experiments in other scenarios, such as reading comprehension, to examine whether the model attends to answer-relevant tokens, which I believe are also important for decision-making.
> >
> > For the second weakness, I appreciate the balance achieved between performance and explainability. I wonder whether the enhanced explainability of B-cos LMs could also contribute to improving performance. For example, prior work on attention-based explanation methods has shown that intervening on the attention distribution can sometimes yield performance gains. If model performance and explainability could be shown to mutually reinforce each other, this would considerably strengthen the contribution and practical significance of your work.

---

> > > ### Author Response · Authors · 2025-09-11
> > >
> > > Thank you for your valuable feedback.
> > >
> > > We fully agree that extending our methods to more general scenarios is important and highlight this as a future direction, once suitable benchmarks for evaluating explanations in such flexible settings are available.
> > >
> > > As discussed in Section 6, B-cosification does not improve task performance. Unlike methods that directly intervene on explanations such as [1], our approach only modifies the model architecture without constraining explanation forms, which may limit its effect on performance. Given the faithfulness of B-cos explanations, we hypothesize that explicitly intervening on them could better regularize model behavior, and we plan to explore this in future work.
> > >
> > > [1] Kennedy et al., Contextualizing Hate Speech Classifiers with Post-hoc Explanation, ACL 2020

---

### Review · Reviewer_PXTs · 2025-08-28

**Summary Of Contributions:**

## Strengths
1. This paper first introduces B-cos netowrks to the language model setting, specifically the classification task and generation task, which help improve the language model's explanability with minimal accuracy scrafice.
2. This paper prevides comprehensive evaluation, including both automatic metrics and human evaluation studies demonstrating improved explanation faithfulness and interpretability compared to post-hoc methods. Also various ablation study setting for B values, and training setups. This provides the community enough insights and refernece in the future developement.

## Weaknesses
1. The selection of B values for different tasks, as shown in the paper, may vary and it usually takes additional experiements to determine the best value that balance the accuracy and explanability. It's will be good to see whether B-value can be also included as a trainable paramtere to dynamicly select during the adaption process.
2. For

**Additional Comments:**

1. How does the approach scale to larger language models (e.g., models with billions of parameters)?
2. How does B-cos LM generalize and work for those long-form generation task or they may only work in the classfication task?

**Audience:**

Yes

**Audience Explanation:**

This paper presents a way to make langauge model's more explanable and really did a comprehensive experiments to validate the sentitiveness of various hyperparameters like B values. This is proven to be a quite efficient way as specified in Table 11, though requring additional fine-tuning to transforming a LM into a B-cos LM. Therefore, this paper first introduce B-cos way into the language model and can be interesting to those people who focus on the langauge model explanability, like explaination to customers, etc.

However, this paper may also be limited in the generation task as all the tasks in the paper are classfication tasks (even for generation models). How does B-cos LM explain those long-form generation tasks is not discussed. In fact, generative LMs may actually self-explain it's response through the long response generated. How to position B-cos LM in generation tasks may be a undetermined topic.

**Broader Impact Concerns:**

As a explanation method, B-cos LM may have a possibility to give wrong explanation and thus mislead the user in understanding the LM's behavior.

**Claims And Evidence:**

Yes

**Claims Explanation:**

Yeah, the claims are supported by comprehensive experiments mentioned in section 4, 5, 6, 7, including auto-metrics like comprehensiveness, sufficiency, and human evaluation, case visualzation, and ablations studies.
For exmaple:
1. The authors cliam B-cos LMs produce more faithful explanations than post-hoc methods. The author prove this by results in Table 2, 8, 9
2. The authors claim that after fine-tuning LM to a B-cos LM, the accuracy is still comparable to conventional methods, this is supported by Figure 2, though there is a little performance drop on IMDB (3 points) compared to the conventional LM method.
3. The authors also claim that it's more efficient in terms of both time and memory occupation, as specified in Table 11.

**Requested Changes:**

1. Although Figure 5 provides detailed ablation study about the B vallue  selection, I think the authors should write a guidance based on the experiments insights about the selection of B-cos LMs under different task setting, so people don't need to do a bunch a ablation studies to determine a good value for them.
2. Are there any failure cases where B-cos LM make explicitly wrong explanation and this can help people understand better about when B-cos LM not work and better understand its behavior.

---

> ### Author Response · Authors · 2025-09-05
> **Reply to Reviewer PXTs**
>
> We thank the reviewer for their time and thoughtful comments. We have carefully considered each point and provided detailed responses, and we hope that our clarifications and revisions address the concerns raised.
>
> # [Weakness 1 & Change 1] Trainable B & Guidance for B selection
>
> We chose not to set B as a trainable parameter for two reasons:
>
> 1. We believe treating B as a tunable hyperparameter gives users explicit control over the trade-off between task performance and explainability, which is an important flexibility in applications where one may be prioritized over the other.
>
> 2. Prior work [1] has shown that training B does not improve task performance or explainability; they therefore recommend treating B as a hyperparameter.
>
> We appreciate the suggestion to provide guidance for selecting B. In the revised version, we will add a dedicated section (before the conclusion) outlining practical strategies for choosing B efficiently.
>
> ```
> Practical Guidance for Training B-cos LMs
>
> Based on our experiments and analyses, we provide the following guidance on configuring B-cos LMs:
>
> B-cosification Setup: For efficient B-cosification of encoder-only models, we recommend combining B-cos conversion and task fine-tuning. However, if resources permit, an additional B-cos pre-training stage can further enhance both task performance and explanation faithfulness. For decoder-only models, B-cosification can be applied to pre-trained models with unsupervised training data, enabling their conversion into B-cos LMs with improved explainability.
>
> Choice of B value: Several factors influence the selection of an appropriate B value:
> (1) Model capacity and explanation sparsity: Excessively large B values can reduce model capacity and produce overly sparse explanations. We therefore recommend setting B within the range of 1–2.
> (2) Task complexity and language diversity: For complex tasks with varied language use (e.g., online forum data with diverse language styles), smaller B values are preferable, as they preserve model flexibility and capture more useful features. For other tasks, larger B values can improve explainability.
> (3) Bias considerations: Be mindful that B-cosification may amplify biases that the model learns from biased training data. When this risk is present, consider choosing smaller B values and applying bias mitigation methods if necessary.
> ```
>
> # [Change 2] Undesired B-cos explanations
>
> We appreciate this valuable suggestion. In the revised version, we include representative examples of undesired B-cos explanations (Figure 13, Appendix K) to illustrate the limitations and provide a more balanced view of the method. Please refer to the revised manuscript for the examples and accompanying analysis.
>
> # [Broader impact concern] Potentially wrong and misleading explanations
>
> While B-cos LMs may still produce undesired explanations, our results demonstrate that their explanations are generally more faithful and more interpretable to humans than those generated by existing methods. We therefore believe that B-cos explanations hold promise for providing more accurate and useful insights into model computations than current alternatives.
>
> # [Comment 1 & 2] Generalization of B-cos LM to larger LLMs and long-form tasks
>
> Our experiments in Section 7 indicate that the B-cos transformation can be applied to larger models to enhance explainability with minimal loss in language abilities. However, we also observe that larger models often require substantially longer fine-tuning to adapt to the B-cos architecture, which can lead to high training costs. Exploring ways to integrate B-cos into the pre-training stage, or to combine it with more efficient adaptation methods, is therefore an important direction for future work. Furthermore, assessing the impact of B-cosification on other capabilities of LLMs (e.g., knowledge retention and reasoning) lies beyond the scope of the present study but remains an open question.
>
> Regarding long-form generation tasks, our results in Section 7 already demonstrate that B-cos LMs can explain predictions in open-ended generation settings (beyond classification). The same approach can be extended to long-form generation by providing token-level explanations. Nonetheless, evaluating explanation quality in such complex scenarios remains a challenge for the community, and token-based input explanations may not always be the most suitable representation. For these reasons, we did not include additional experiments on long-form generation in this work.
>
> # Literature
>
> [1] Arya et al., B-cosification: Transforming Deep Neural Networks to be Inherently Interpretable, Neurips 2024
>
> **Besides, we noticed that Weakness 2 was mentioned in the summary but not detailed in the review. If there are additional concerns corresponding to this point, we would be happy to address them in detail.**

---

### Author Response · Authors · 2025-09-05
**Reply to Action Editor and All Reviewers**

We sincerely thank the action editor and all reviewers for their time and effort in providing valuable feedback. We have carefully revised the manuscript to address the concerns raised. The main changes are as follows:

1. We moved Section 3.2 (Dynamic Linear Representation of Model Components) to Appendix C in order to streamline the main text.

2. We added a dedicated Section 8 (Practical Guidance for Training B-cos LMs), which provides recommendations on how to configure B-cos LMs and choose appropriate B values based on our findings.

3. We expanded the discussion in the limitations section to clarify when B-cos explanations are suitable and when they may be less effective.

4. We included two representative examples of undesired effects of B-cos explanations (Figure 13, Appendix K) to provide a more balanced view of the method.

For further details, please refer to the revised manuscript and our replies to each reviewer. We once again thank the AE and reviewers for their constructive comments and suggestions.

---

### Decision · Action_Editor_JvnX · 2025-11-05

**Recommendation:** Accept as is

**Audience:**

Yes

**Audience Explanation:**

Researchers who work on model interpretability.

**Claims And Evidence:**

Yes

**Claims Explanation:**

The paper introduces B-cos LMs, which convert pretrained language models into bias-free dynamic linear models to make explanations faithful and interpretable. They remove all bias terms, replace linear layers with a cosine-alignment B-cos transformation, drop non-linear activations in classification heads, and fine-tune. The automatic and human evaluations show that B-cos LMs deliver explanations that are more faithful and more interpretable to humans than post-hoc methods, while maintaining task performance on par with conventional fine-tuning.

Reviewers noted that the technical novelty is limited but well executed, and that readers can still benefit from this paper.